# Deep phenotyping unveils hidden traits and genetic relations in subtle mutants

Adriana San-Miguel[1,†], Peri T. Kurshan[2], Matthew M. Crane[3,†], Yuehui Zhao[4], Patrick T. McGrath[4], Kang Shen[2] & Hang Lu[1,3]

Discovering mechanistic insights from phenotypic information is critical for the understanding of biological processes. For model organisms, unlike in cell culture, this is currently bottlenecked by the non-quantitative nature and perceptive biases of human observations, and the limited number of reporters that can be simultaneously incorporated in live animals. An additional challenge is that isogenic populations exhibit significant phenotypic heterogeneity. These difficulties limit genetic approaches to many biological questions. To overcome these bottlenecks, we developed tools to extract complex phenotypic traits from images of fluorescently labelled subcellular landmarks, using *C. elegans* synapses as a test case. By population-wide comparisons, we identified subtle but relevant differences inaccessible to subjective conceptualization. Furthermore, the models generated testable hypotheses of how individual alleles relate to known mechanisms or belong to new pathways. We show that our model not only recapitulates current knowledge in synaptic patterning but also identifies novel alleles overlooked by traditional methods.

[1] School of Chemical and Biomolecular Engineering, Georgia Institute of Technology, Atlanta, Georgia 30332, USA. [2] Department of Biology, Howard Hughes Medical Institute, Stanford University, Stanford, California 94305, USA. [3] Interdisciplinary Program in Bioengineering, Georgia Institute of Technology, Atlanta, Georgia 30332, USA. [4] School of Biological Sciences, Georgia Institute of Technology, Atlanta, Georgia 30332, USA. † Present addresses: Department of Chemical and Biomolecular Engineering, North Carolina State University, Raleigh, North Carolina 27606, USA (A.S.-M.); Department of Pathology, University of Washington, Seattle, Washington 98195, USA (M.M.C.). Correspondence and requests for materials should be addressed to H.L. (email: hang.lu@gatech.edu).

Understanding gene function and the structure of biological networks relies heavily on identifying morphological, functional or behavioural phenotypic changes upon genetic perturbations. With deep-sequencing and the recent advances in genome-editing tools, the bottleneck for discovery of cellular functions now is phenotypic analyses[1–5]. Most genetic studies thus far focus on mutations causing dramatic phenotypic differences that can be easily assessed by qualitative visual inspection. Yet, an increased interest in understanding genetic mechanisms of population heterogeneity[6], how noise affects network responses[7,8] and how developmental systems compensate for perturbations[9,10] means that simplistic, qualitative phenotypes are insufficient. In addition, many biological perturbations relevant to human diseases have subtle phenotypes not necessarily accessible to eye[1,11,12]. This challenge may be addressed, in part, by sensitive, reliable methods to robustly characterize subtle phenotypes.

While image-processing tools have been increasingly applied to perform quantitative analysis, this has been mostly useful for *in vitro* cell models[13–23]. For live *in vivo* models (for example, genetic organisms), the number of markers (for example, fluorescent reporters) that can be simultaneously used is usually small, thus limiting the dimensions of the phenotype to be scored. There is also inherent complexity of working with intact animals, for instance, forward genetic screens in small model organisms are usually performed by phenotyping single animals, not populations of clones as in cell culture. These difficulties have prevented extensive use of large-scale high-resolution image-based studies. In *Caenorhabditis elegans*, live phenotyping has mostly focused on drastic changes exhibited on gross features (for example, whole-animal or tissue-level changes)[24–26]. Most applications involving end point high-resolution imaging are low dimensional, with the exception of live tracing of cell lineages and quantification of gene expression in embryos[15,27,28]. However, the rich information encoded in fluorescence images of multicellular models has not been fully exploited at high resolution (that is, characterization of subcellular features within a living multicellular organism, thus missing the identification and characterization of phenotypic changes of weak alleles[29,30]). Although different approaches have been used to identify chemically or genetically induced phenotypes, these have typically either screened for severe changes or have focused on behavioural or anatomical changes[5,31–38].

One particularly challenging area for quantitative phenotypic profiling is synaptic patterning in *C. elegans* as a test case for the general strategy. Here we present an approach to perform comprehensive multidimensional phenotypic profiling using fluorescently labelled synaptic puncta in *C. elegans*. Our combined computational and experimental approach addresses the major challenges to isolate and characterize subtle alleles by quantifying micron-sized subcellular landmarks in an unbiased manner from a single genetically encoded reporter. This integrative approach is based on applying statistical methods to capture the phenotypic heterogeneity in isogenic populations, creating phenotypic profiles containing multiple and complex features, and developing models to place subtle alleles in the phenotypic space. We demonstrate an approach to identify genetic alterations that give rise to subtle and varying phenotypic changes, unintuitive to and difficult to assess by human perception, but nonetheless relevant in the nematode *C. elegans*. Because of the large dimensions of the phenospace available to the computational approach, we can now identify changes from an ensemble of phenotype characteristics, use them to cluster alleles and give rise to new hypotheses of genetic relations. Using this method, we identified a new allele with a surprising role in *C. elegans* synapse morphology and function.

## Results

**Deep phenotyping reveals isogenic population heterogeneity.** In *C. elegans*, synaptic patterning is thought to be stereotypical[39–42], and the use of fluorescent reporters has enabled the discovery of many genes involved in synaptogenesis[43]. We focus on synaptic patterns at the neuromuscular junction, specifically in the DA9 motor neuron (Fig. 1a and Supplementary Fig. 1a). Genes involved in synaptic assembly have been identified by screening grossly mislocalized presynaptic material[42,44–49]. Identifying weak alleles (for example, small differences in size, intensity or distribution of micron-sized puncta as compared with those of wild type) is an arduous (sometimes impossible) undertaking. Weak alleles might also exhibit pleiotropy with a variety of phenotypic characteristics affected, many likely imperceptible to or difficult to assess by human vision. The challenge resides in accurately quantifying metrics that fully capture image-based phenotypes, and in identifying the relevant differences that characterize subtle alleles (Fig. 1b,c).

Using a modified large-scale imaging and sorting system[45,50–53] (Supplementary Fig. 1 and Supplementary Note 1), we performed multidimensional profiling of the synaptic puncta in DA9. We developed a microfluidic device capable of orienting animals for optimal imaging of the axon of DA9, located on the dorsal side of the animal. Puncta identification was performed based on support vector machines, while quantitative descriptive metrics of axonal synaptic patterning were extracted for characteristics that are difficult to assess by visual inspection. While previous phenotyping of synaptic patterning in DA9 has mostly focused on localization defects[42–44,46,47,49,54,55], here we focus on metrics important to identify much subtler phenotypes, such as the size, intensity and homogeneity of synaptic puncta. The multidimensional phenotypic profiles incorporate information from a total of 76 descriptive different metrics. We developed a data analysis pipeline for automated image annotation (Supplementary Figs 1–3 and Supplementary Note 1), and identification of the characteristics that differentiate mutant phenotypes from wild type. Contrary to the conventional notion of the stereotyped development[39–42], isogenic populations of wild type and previously isolated strong mutants exhibit surprisingly significant heterogeneity in various phenotypic descriptors even under strictly controlled culture conditions. These descriptors include biologically important measures such as puncta size, number and intensity, and the synaptic domain length (Fig. 1d,e). While means of these metrics may be different between mutants and wild type, many are obscured by the variance in the data. Even a comparison of the most distinct metrics is unable to perfectly separate the mutant and wild-type populations (Fig. 1f). These data suggest that it is in general impossible to establish an unequivocal threshold to separate populations using single features (as illustrated in Fig. 1c). Therefore, previous screens that identified mutants focusing on one trait (such as synaptic localization)[56,57], or based on qualitative inspection, must have relied on favourable sampling of these populations (either by having screened more than one animal from these genotypes or by having stochastically sampled a phenotypically severe individual); in other words, many alleles could have been missed in these screens, and even in the screens for severe phenotypic changes.

**Subtle alleles exhibit complex multidimensional phenotypes.** To identify new alleles that affect morphology, particularly those that may be subtle or of previously unidentified phenotypes, we conducted an automated unbiased forward genetic screen (Supplementary Note 2). Because the screen is not directed towards specific phenotypes, we chose to set the sorting threshold in phenospace where the occurrence of a wild type is rare.

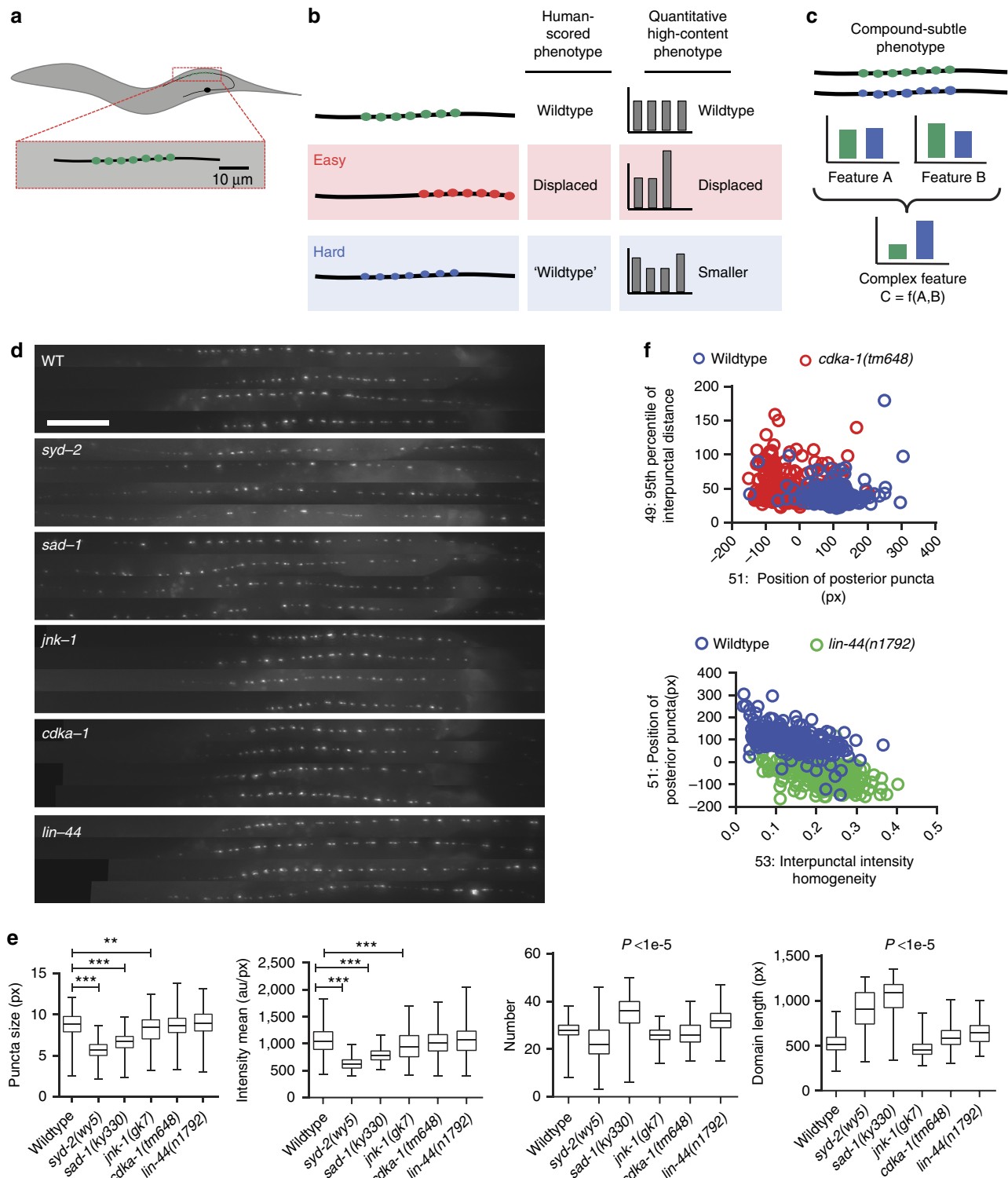

**Figure 1 | Identification of subtle phenotypes requires unbiased quantitative information.** (**a**) Schematic representations of presynaptic sites at the DA9 motor neuron axon. (**b**) Phenotypes imperceptible by qualitative analysis can be detected by unbiased computation of multiple descriptors. (**c**) Computation of complex features enables the identification of non-obvious alleles. (**d**) Typical fluorescence micrographs of synaptic patterns (four images per genotype) for wild-type and some mutant genotypes; scale bar, 20 μm. (**e**) Boxplots showing heterogeneity in synaptic descriptors from populations of wild-type and known mutants. Box represents 25th, 50th and 75th percentiles; whiskers show minimum and maximum. (**f**) The two features with means most different from wild type for *cdka-1* (top) and *lin-44* (bottom); even descriptors with significant differences exhibit data overlap between wild type and mutant. N = 443 (wild type), 124 (*cdka-1(tm648)*), 225 (*lin-44(n1792)*), 239 (*syd-2(wy5)*), 126 (*sad-1(ky330)*) and 152 (*jnk-1(gk7)*). \*\**P* < 0.001 \*\*\**P* < 0.0001. *P* values in plots for number and domain length are for comparisons of all groups versus wild type. *P* values obtained from a multiple comparison test (Kruskal–Wallis), with a 99% confidence level and Bonferroni correction. Statistical values displayed only for comparisons versus wild type.

Specifically, animals were positively sorted if any of the phenotypic features exceeded a pre-set threshold. Because of the stochasticity in the phenotypes, this step serves as an enrichment for true mutants but cannot definitively identify them. In total, we screened ~4,000 haploid genomes and sorted 155 worms (~3.7%; Supplementary Fig. 4). Viable and nonsterile animals were ranked based on the initial phenotypic scores, 24 of which were subjected for further analysis (Supplementary Note 2).

To determine whether the mutants breed true and how they are different from wild type, we next characterized the phenotypes of isogenic populations of the putative mutants (Supplementary Fig. 4). Because mutants were isolated based on differences from a few specific parameters and, only from a single instance of phenotyping, the exact nature (which is highly unlikely to be the same as the initial sorting criterion) and magnitude of the phenotypic alterations are unknown. We thus conducted comprehensive image-based phenotypic profiling (76 metrics, Supplementary Note 1) of wild type and all mutant populations. We also included previously identified mutants as part of the study and validation. Our analysis contains a total of 41 lines (16 previously identified mutants, 24 isolated in this screen and wild type), for a total of ~6,000 images

(Supplementary Fig. 5). To answer the question to what extent (degree of separation) and in what way (distinguishing features) mutants are different from wild type, we next performed pairwise comparisons between the mutant and the wild-type strains using stepwise logistic regression (SWLR; Fig. 2). For each model, the linear combination of all relevant information represents the mutant phenotypic dimension, and can be used to analyse both strong and weak mutants. The position of each animal on this curve is a predictor of the animal exhibiting the specific mutant phenotype. Figure 2a–c illustrates one visually apparent mutant (arl-8) and one subtle mutant found from the screen (a178). For arl-8, a visually identifiable phenotype, the logistic regression model resembles a step function, perfectly separating the mutant from wild type (Fig. 2b), and several of the features by themselves can already differentiate the mutant (Fig. 2c). For weak alleles such as a178, with no obvious phenotype from visual inspection (Fig. 2a), the differences are still evident in the model when compared with wild type along the a178 phenospace (Fig. 2b,c). Owing to the subtlety of the phenotype, it is necessary to integrate information from a large population, rather than from phenotypic analysis of a single animal. From the analysis, a178 has a lower variability in puncta brightness (feature 21), while

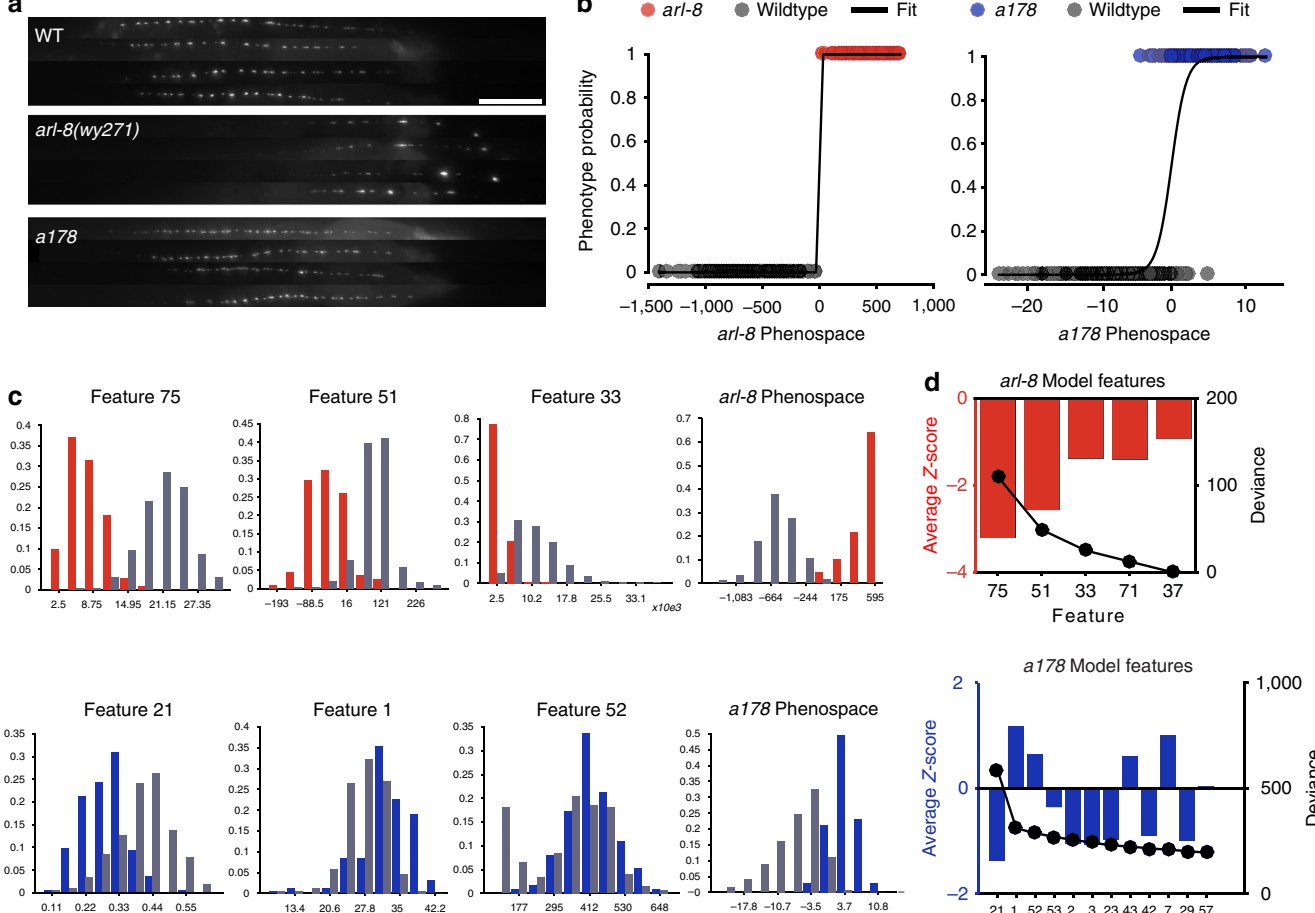

**Figure 2 | Phenotypic differences between populations are revealed by logistic regression models.** (**a**) Images of wild type, arl-8 (easily identifiable) and a178 (no obvious phenotype), four images per genotype; scale bar, 20 μm. (**b**) Plots showing logistic regression fits for obvious (left, arl-8) and subtle (right, a178) phenotypes. (**c**) Histograms for the three most relevant features, and the score in the arl-8 and a178 phenotypic scales. N = 111 (arl-8(wy271)), 228 (a178). Features listed are: 75: number of puncta in the top 75% of size range; 55: 10th percentile of integrated intensity; 33: intensity of posterior puncta; 21: puncta intensity homogeneity; 1: number of puncta; 52: interpunctal intensity. (**d**) Relevant features obtained from stepwise logistic regression (from most to least relevant, left to right) for arl-8 and a178. Black line (quantification on the right axis) shows the change in model deviance as each feature is added during model construction. Full feature descriptions, statistical analysis for model creation and feature selection are listed in Supplementary Information.

also having a larger number of puncta (feature 1), although these two features alone are insufficient to describe how *a178* differs from wild type (Fig. 2c,d). The most significant feature selected for *a178* is a statistical feature (variability), which will certainly elude the eye (Fig. 2a). One advantage of building logistic regression models with forward–backward feature selection algorithm is that only those variables that significantly contribute to the model to differentiate the mutant from wild type are included; it is not surprising that some mutants require many descriptors while others not (Fig. 2c, Supplementary Fig. 6 and Supplementary Notes 3 and 4). Interestingly, the number of

features necessary to distinguish a mutant varies, and is not correlated with the separation efficiency achieved (Supplementary Fig. 7).

We next examined the discriminatory power of each pairwise model for previously identified mutants and for those isolated in our screens (Fig. 3a–c). Overlapping probability plots (Fig. 3a) and slow rising receiver operating characteristic curves (Fig. 3c) with a small area under the curve (AUC; Fig. 3b), for example, *a221*, indicate poor separation of genotypes in the phenospace, that is, mutant having a subtle phenotype. Models for most of the previously identified mutant collection display

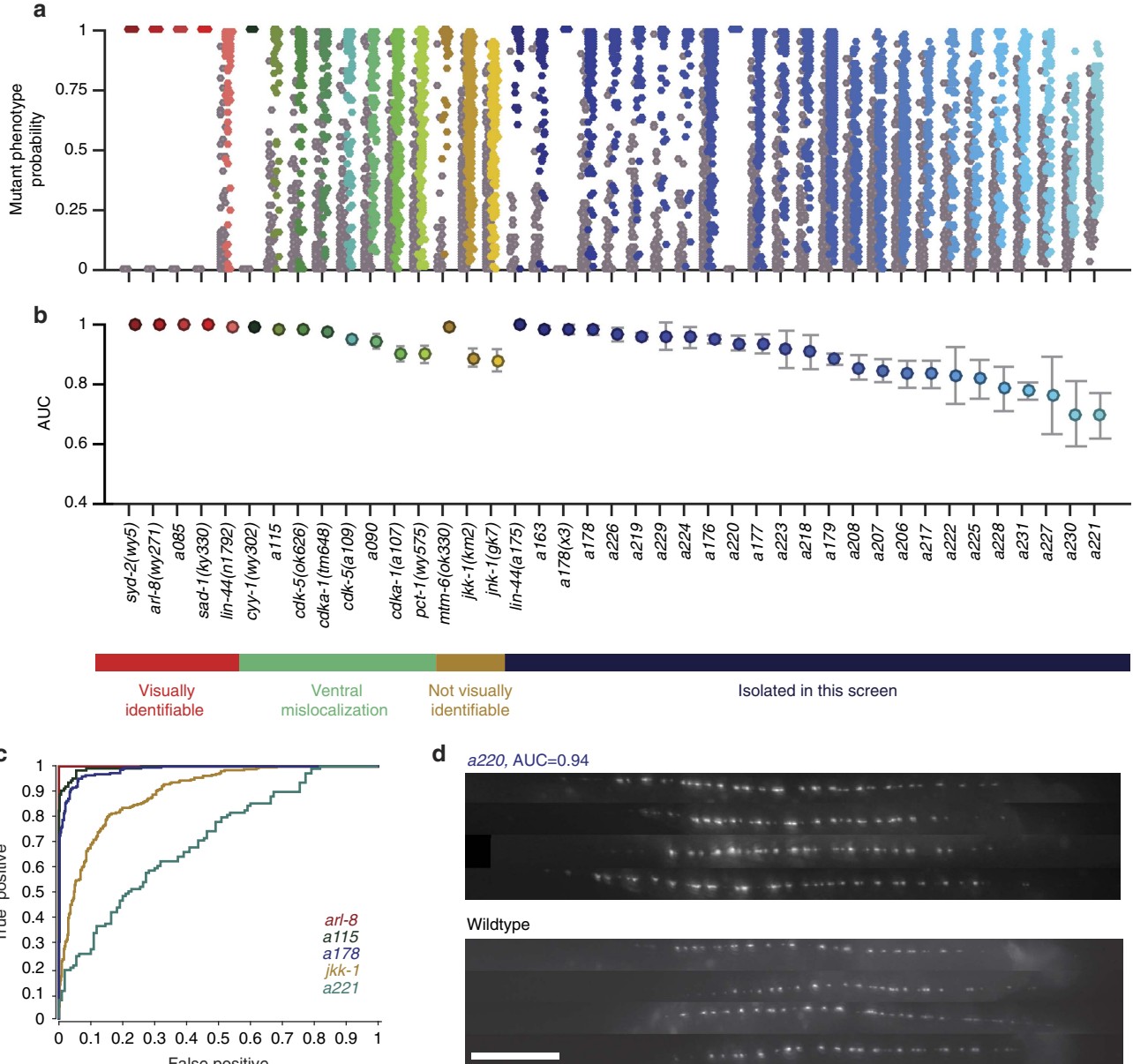

**Figure 3 | Logistic regression models reveal the degree of separation between mutant and wild-type populations.** (**a**) Probability in the mutant phenospace for each animal in the wild type and mutant populations. Each column is scored by a different logistic regression model. Populations are divided into groups, and ranked by severity. (**b**) AUC for each logistic regression model is an indicator of the discriminating power of each model. AUC was quantified in a fivefold cross-validation. Error bars represent s.d. of the AUC form each testing fold. Genotypes are ordered by groups (visually identifiable mutants, ventral mislocalization mutants, not-visually identifiable mutants and mutants isolated in this screen). In each group, strains are ranked from most identifiable to less identifiable, as measured by AUC. (**c**) Receiver operating characteristic (ROC) curve for five populations with varying degrees of phenotypic severity captures the degree of separation by sweeping a discriminating threshold on the computed phenotype probability. (**d**) Images for mutant *a220*, which shows no obvious phenotype but is quantitatively distinguishable from wild type, and for wild type. Scale bar, 10 μm. Sample sizes, statistical analysis for model creation and feature selection are listed in Supplementary Information.

excellent discrimination power (AUC > 0.88), especially for classical visually apparent mutants (AUC > 0.99; Fig. 3a). This includes even the ventral mislocalization mutants (including *cyy-1(wy302), pct-1(wy575), cdk-5(ok626), cdka-1(tm648), a090* and *a115*), which is somewhat surprising since we did not image the ventral side of the worms; this suggests that the amount of synaptic material and their distribution are likely controlled, such that ventral mislocalization affects dorsal synapse morphology. For these animals, visual inspection of the dorsal axon typically cannot differentiate these animals from wild type (Supplementary Fig. 5). Nonetheless, our models handle these mutants well, and report true differences in subtle mutants that would have been overlooked by qualitative assessment. In addition, some mutants previously not identifiable by eye can also be scored by our method: *jnk-1(gk7)* and *jkk-1(km2)*. These were previously identified from suppressor screens of *arl-8* (ref. 58) and show some degree of separation, while *mtm-6(ok330)* (ref. 59) is perfectly separable from wild type (AUC = 0.996).

For mutants from our screen, the phenotype severity varies as expected for weak alleles with large phenotypic heterogeneity. Except for *a175*, which has a posteriorly shifted synaptic pattern, none of the isolated mutants display a visually apparent phenotype. According to logistic regression models, however, some (for example, *a220, a163*) are as easy to identify as the known mutant collection, and others are much more subtle (for example, *a228*), likely because of two reasons: the model's ability to take into account multiple phenotypic features, and

features that are statistical in nature (such as variability), both changes difficult to assess by human perception. Notably, the model for *a220* can almost perfectly discern between mutant and wild-type populations (AUC = 0.94), although visual inspection detects no obvious defects (Fig. 3d).

**Subtle mutant *a178* shows unexpected behavioural defects.** Next, we selected one mutant, *a178*, for further study since it possesses an intriguing composite phenotype, one aspect of which involves reduced variability of synapses within each individual (Fig. 2). Although robust (that is, measurable and reproducible) and stable even after multiple outcrosses, the *a178* phenotype is also extremely subtle to human perception. Moreover, this mutant displayed no obvious defects in standard locomotory assays (Fig. 4a), and seemed normal in aldicarb sensitivity, which is a proxy for acetylcholine neurotransmission[60] (Fig. 4b). We mapped *a178* to a region of chromosome III, which contained a splice site mutation in *sax-2* as determined by whole-genome sequencing (Supplementary Note 5); furthermore, two known alleles of *sax-2* (*ky216* and *ot10*) also phenocopied *a178* using both the models for the original isolate and the three times outcrossed population (Fig. 4c), corroborating that *a178* is an allele of *sax-2*. Interestingly, *sax-2* is previously known to be involved in neuronal morphogenesis[61]. The *a178* model suggests that this population on average has a higher number of puncta with more homogeneous brightness. Thus, we hypothesized that the gene mutated in *a178* may contribute to synaptic material distribution and synaptic function. As compared with synaptic

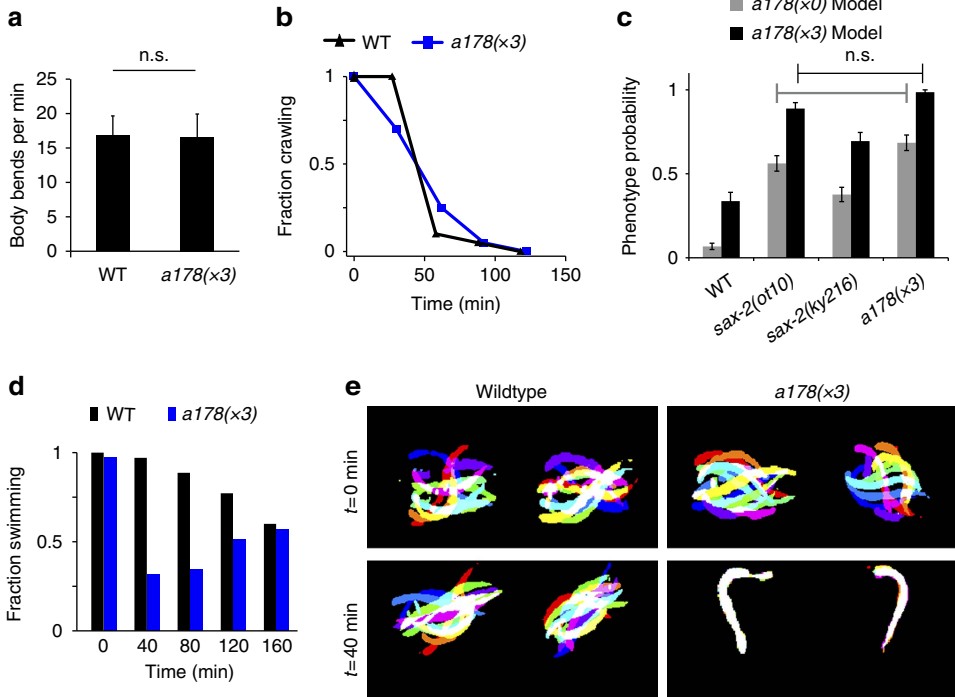

**Figure 4 | Subtle mutant *a178* exhibits robust behavioural phenotypes.** (**a**) Animals crawling on plate show no locomotive differences (ventral body bends per min), *n* = 20 per genotype. Statistical comparison performed by *t*-test with a 95% confidence level. Error bars are standard deviation. (**b**) *a178* exhibits normal acetylcholine release according to an aldicarb sensitivity assay. Plot represents fraction of animals moving after transfer to a 1 mM aldicarb NGM plate, *n* = 20 per genotype. Aldicarb assays replicated five times with no differences between genotypes. (**c**) *sax-2* alleles *ot10* and *ky216* phenocopy the *a178* phenotype. Bar plots represent the average phenotype probability of each imaged population, according to the SWLR models built for *a178(x0)* and *a178(x3)*. Error bars are s.e.m. Statistical comparison performed by *t*-test with a 95% confidence level and Bonferroni correction for multiple comparisons. Only nonsignificant differences displayed, all other comparisons have *P* < 1e − 6. (**d**) *a178* animals display a drastic time-dependent reduction in swimming. Plot represents fraction of animals swimming after transfer to liquid from plate, *n* = 35 per genotype. Replicate experiment with *sax-2* alleles in Supplementary Fig. 8. (**e**) Swimming animals represented as an overlay of segmented images (10 frames, 750 ms, see Supplementary Videos 1–4), two individual worms per genotype and time point.

patterning mutants (such as *lin-44* or *syd-2*), *a178* exhibits a much subtler phenotype. Without confirmation of a functional defect, it would be difficult to positively argue that *a178*, although morphologically different from wild type, is a relevant mutation that affects synaptic function. Notably, posterior to isolation, we identified a drastic behavioural phenotype exhibited by *a178*. These animals display a distinct swimming phenotype when transferred to liquid media. Although no difference was observed in initial swimming speed, *a178* reduces thrashing frequency drastically over time (Fig. 4d,e and Supplementary Videos 1–4) as compared with wild type. The phenotype of *a178* resembles the swimming-induced paralysis previously observed in some mutants (for example, *dat-1*)[62] with defective transmitter reuptake, which suggests that *a178* could be defective in rates of neurotransmitter release or reuptake while maintaining rapid and sustained neuronal activation rates. The two *sax-2* alleles also reduce swimming locomotion faster than wild type, although to a smaller extent (Supplementary Fig. 8). The results suggest that the subtle *a178* synaptic patterning phenotype may be related to inefficient neurotransmission, particularly in vigorous activities that would require fast rates of neurotransmitter release and recycling.

**Multidimensional phenotype models are specific and sensitive**. We next characterized the multidimensional phenotype models' sensitivity (how well each model picks up all animals in each population) and specificity (how well the model can identify wild-type animals as such). Because they were built to differentiate the mutants from wild-type animals, the model can be used to classify animals in a binary manner, but can be tuned to meet different false-positive or false-negative criteria (as the receiver operating characteristic curves show, Fig. 3). In order to compare all models, we quantified the sensitivity (true-positive rate) and specificity (true-negative rate) while applying a threshold probability of 50%. To characterize sensitivity, we computed the fraction of mutants picked up by the multivariate SWLR as compared with that of the best feature (BF) models. We apply the BF model as a best-case scenario when using single metrics to characterize the mutants. In reality, and without a prior feature selection algorithm (like the one performed here through SWLR), the results of the BF model are an optimistic estimate of the results when using a single-parameter approach (Fig. 5a top). Not surprisingly, SWLR shows a much higher sensitivity than BF. On the other hand, it is evident that increasing the complexity of the models only moderately improves specificity (Fig. 5a bottom), although this could be because BF models themselves already reach high values of specificity. An increase in model complexity could possibly entail overfitting, describing noise rather than actual phenotypes and resulting in reduced specificity. To test whether this trade-off exists with SWLR, we computed the fivefold cross-validated accuracy for BF and SWLR models (Supplementary Fig. 9), which show that increasing model complexity significantly improves model fit, without compromising model performance. SWLR effectively overcomes the oversimplified descriptions of the phenotypes in BF. Thus, unlike classification schemes based on individual features, SWLR does not have to sacrifice sensitivity for specificity, or *vice versa* (Fig. 5a). As expected, specificity correlates with the severity of the phenotype; models for more obvious mutants perform better (higher AUCs, higher sensitivity), and are also more specific.

We next asked whether the *sax-2 (a178)* phenotype is specific. It is conceivable that the model to describe a weak phenotype (meaning where the difference between mutant and wild type is small) may pick up many other genotypes, and would indicate

that the weak phenotype may be describing noise in the measurements instead. When applying the SWLR model to images from animals of all other genotypes, none other than the original isolate and the three times outcrossed strain had a probably higher than 0.5, indicating that the model is specific against all other genotypes (Fig. 5b and Supplementary Note 5). For comparison, the *sax-2* alleles, *ot10* and *ky216*, also scored above 0.5 (Fig. 4c).

To ask how well the models we developed in this study represent the specific mutants rather than representing mutants in general, we next examined the global performance of all 40 SWLR models in a pairwise validation study with all genotypes (Fig. 5c). As designed, wild type has very low probability of being identified as a mutant by any of the mutant models, shown as the small grey dots on the top row. Also as designed, the models pick up the corresponding mutants effectively, as shown by the large colour dots on the diagonal line. Severe mutants (rows whose legends are highlighted in green) appear to be identifiable by a significant number of models; it is likely that these mutants are severely defective in a majority of the metrics measured, and thus models that include these features (which are the majority of the models) are able to identify them. On the other hand, models for these severe mutants (columns whose legends are highlighted in purple) are more selective than other models, as they pick up few other mutants; this is because they likely contain fewer and more dramatic descriptors, and thus are less promiscuous. For instance, the visually apparent mutant *arl-8* can be detected by a large number of models (row highlighted in yellow), whereas the *arl-8* model is extremely selective. In contrast, subtle mutants isolated in this work are mostly only identified by their phenotype-specific models. For instance, *a178* is exclusively identified by its own model and by the model for *a178(x3)*, and *vice versa* (Fig. 5c,d).

Because promiscuous models suggest that high-probability mutants using these models may share a set of features in the phenospace, we next explored the information contained in these model–data relationships. In Fig. 5e, each edge represents a best-performing model–population pair (besides the population used to build the model). As shown in this figure, relationships between related phenotypes are present: *a178* connects with *a178(x3)*, *a176* connects with *a176-R2* (an independent imaging set of *a176*); similarly, the two alleles for *lin-44* (including *a175* isolated here) are linked; mutants *cyy-1*, *cdka-1*, *cdk-5* and *pct-1*, which are all part of parallel pathways that regulate targeting of presynaptic components to the axon[47], are part of the same subnetwork, which also contains a new allele *a218*, suggesting that *a218* may participate in the same pathway of synapse elimination and formation.

**Visualization tools aid in recognition of altered phenotypes**. While the information provided by the significant features can help interpret the biological defects of the mutants, the multi-dimensional descriptors and 'meta-features' can be difficult to conceptualize. We therefore built a visualization tool to illustrate the average phenotype for each population (Fig. 6). Features included in the schematics are those that can easily be represented, such as population averages of synapse size and intensity, interpunctal distance and location of the most posterior synapse (Supplementary Note 6), while other complex features are not included. Synapses are shown in descending order according to size and intensity. For example, comparing *a178* and wild type, we see that *a178* shows a lower variability in puncta brightness (more homogeneous colouring), in addition to a slightly larger number of synaptic sites; this is difficult to articulate when just looking at raw images of both populations (Fig. 2a) as noted earlier. Alternatively,

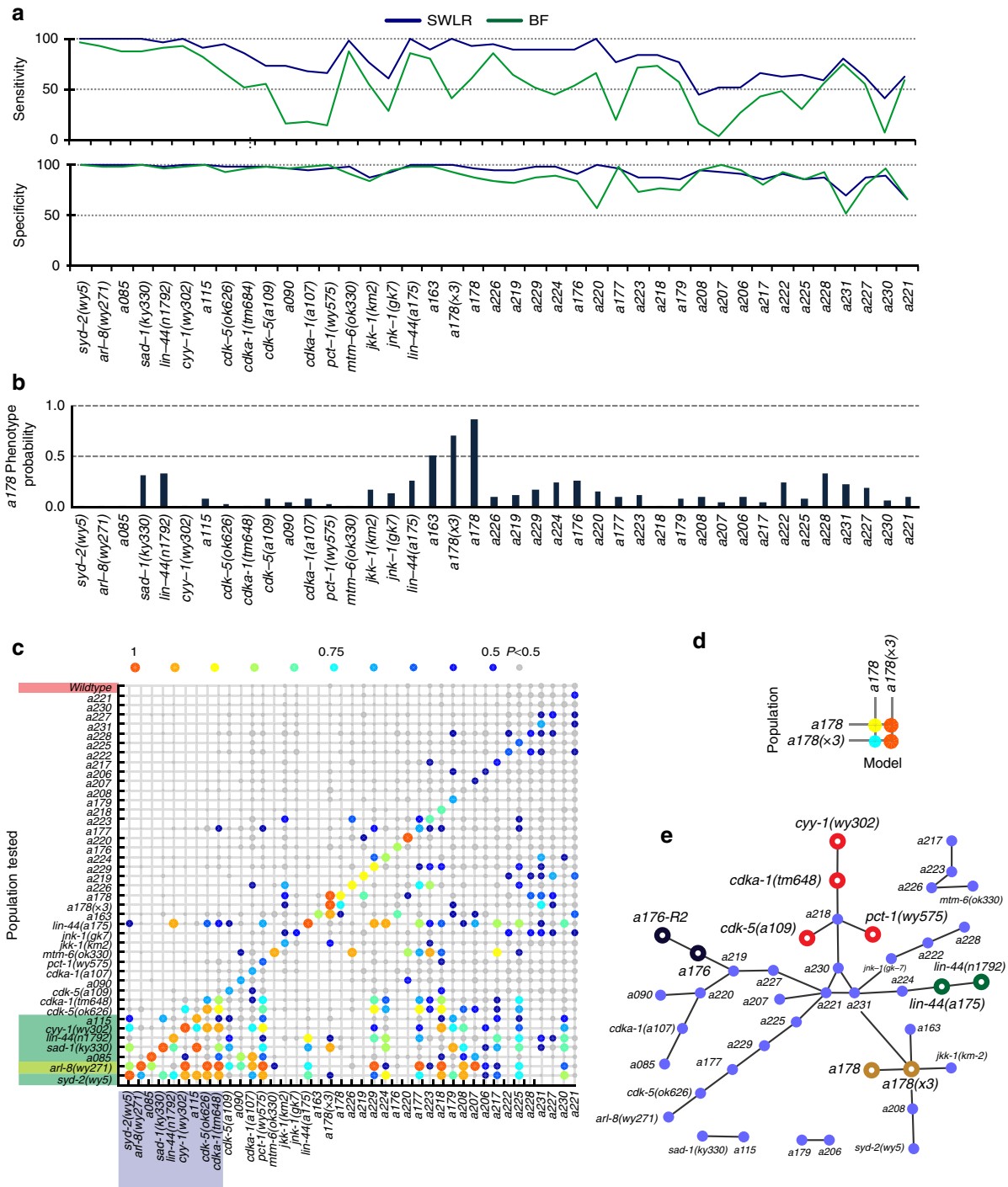

**Figure 5 | Multidimensional models are sensitive and phenotype-specific.** (**a**) Sensitivity (top) and specificity (bottom) of each SWLR and BF model. SWLR models are more sensitive and more specific than the best possible single-feature model. Genotypes are ordered from most identifiable to less identifiable, from each group (refer to Fig. 3a). (**b**) SWLR model for *a178* distinguishes only *a178* populations, and barely population *a163*. Bar plots represent the average *a178* average phenotypic probability of each population. (**c**) Matrix representing the performance of each SWLR model and the detectability of each mutant under other models. Coloured dots represent cases where the specific tested population (rows) scored with an average phenotypic probability above 0.5 under the tested model (columns). The size of the dot, as well as its colour, represents the average probability resulting from each model. Grey dots represent an average probability below 0.5. Dots plotted on the diagonal line show that each model is able to identify its own population with a probability > 0.5 (except for two very subtle mutants), while the probabilities for wild type are all below 0.5 (no highlighting). Rows highlighted in green are severe mutant populations. Columns highlighted in purple are models for severe mutants. (**d**) Models for *a178* detect *a178* populations. (**e**) Identification of the population that each model best identifies (besides itself) reveals phenotypic similarities and corroborates known relationships. Empty circles represent cases where some genetic or phenotypic relationship is known. Coloured nodes are instances where known genetic relationships are confirmed in the network analysis. (*a176-R2* is a different imaging set from *a176*, several generations later, *a178(x3)* is a three times outcrossed *a178*). Sample sizes, statistical analysis for model creation and feature selection are listed in Supplementary Information.

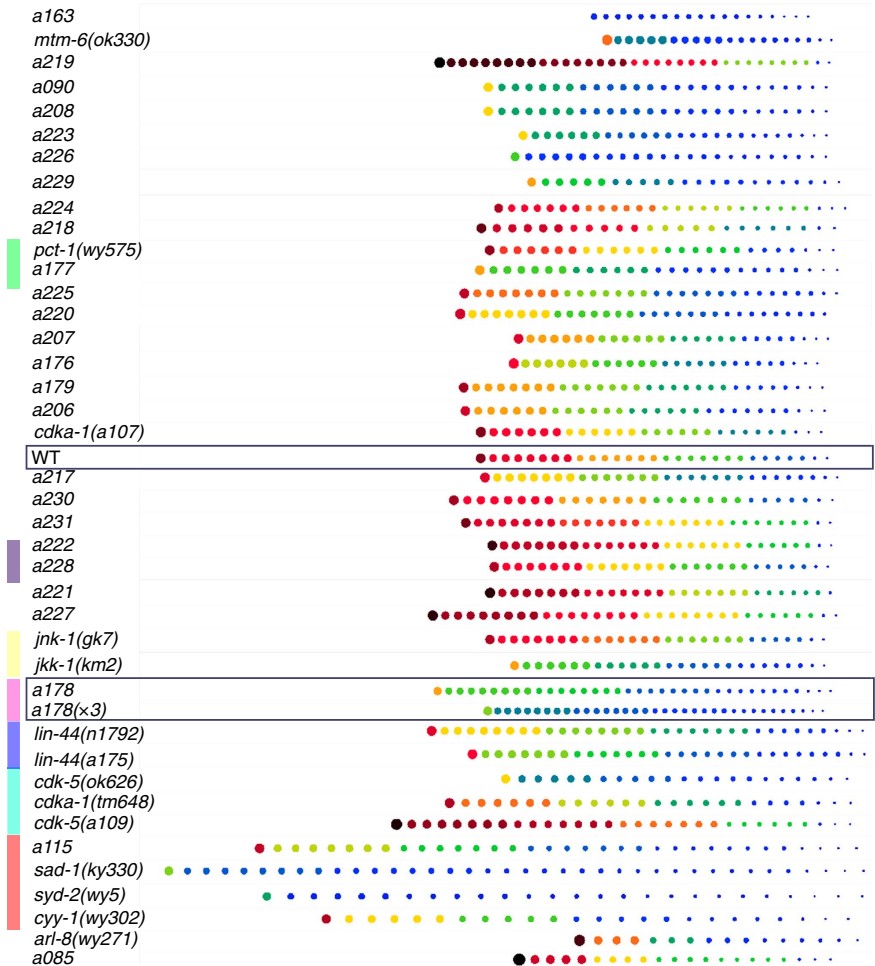

**Figure 6 | Visualization tools aid in the identification of phenotypic changes.** Schematic of the dorsal synaptic puncta of the 'average worm'. Schematics were constructed based on size distribution, interpunctal distance, intensity distribution and number of puncta. Synapses are arranged from largest to smallest, and from brightest to dimmest for visualization purposes. Colour represents intensity. Genotypes are arranged and highlighted according to Fig 7a.

we categorized defects and illustrate the importance and magnitude of changes in another representation to facilitate formulating hypotheses of gene functions (Supplementary Fig. 10; Supplementary Notes 9 and 10).

**Allele severity and gene-relationship spectrum.** Quantification of phenotypic similarity has proven useful at dissecting genetic relationships in studies of embryogenesis, synaptic transmission, lifespan and insulin signalling, and the role of essential genes[14,15,55,63,64]. We next asked whether the high-resolution phenotypic profiles can be used to place the subtle mutants into known pathways and thereby help to define the mutants' functions. Since the phenotypic changes are subtle and present in multiple features, and may include complex 'meta-features', manually curating these alleles and placing them in putative pathways is impossible. We thus hypothesized that compiling information from the full phenotypic profile and comparing it to phenotypes of known mutants could lead to improved theories of gene function. Here we use hierarchical clustering to correlate the full spectrum of phenotypic alterations in the isolated mutants to those of the known mutants. Distances between genotypes in the phenospace were calculated based on the average $z$-scored phenotypic profiles for every population with all 76 features (Supplementary Notes 3, 7 and 8). Since the features that make up

the phenotypic profile are not independent, a standardized Euclidean distance, weighted by the inverse of the correlation coefficient, was used. Thus, all features were taken into account while avoiding biases towards aspects more heavily represented in the phenotypic profile.

Hierarchical clustering recapitulates known relationships between genotypes, as well as predicting novel relationships (Fig. 7a). Further, the same information (of distances between genotypes) is represented visually in the phenospace as a network, which highlights the dramatic mutants found in conventional screens as outer vertices and those in clusters near the wild type as the weak allele found in our screens (Fig. 7b). In general, phenotypes span a spectrum, with closeness suggesting functional relatedness. For instance, genes previously known to work on synaptic vesicle clustering at presynaptic terminals[46,65–67], sad-1 and syd-2, do cluster together. Another example is allele a175, which clustered with lin-44, suggesting that a175 could be involved in the same Wnt signalling pathway[49]. We subsequently tested this hypothesis and indeed found that a175 is an allele of lin-44 by conventional complementation tests, where phenotypic analysis was performed by visual inspection (since a175 is a very easily identifiable phenotype).

Remarkably, mutants jnk-1 and jkk-1, each visually indistinguishable to wild type, also cluster together. These genes are known to work in the same pathway: JNK-1 activity is

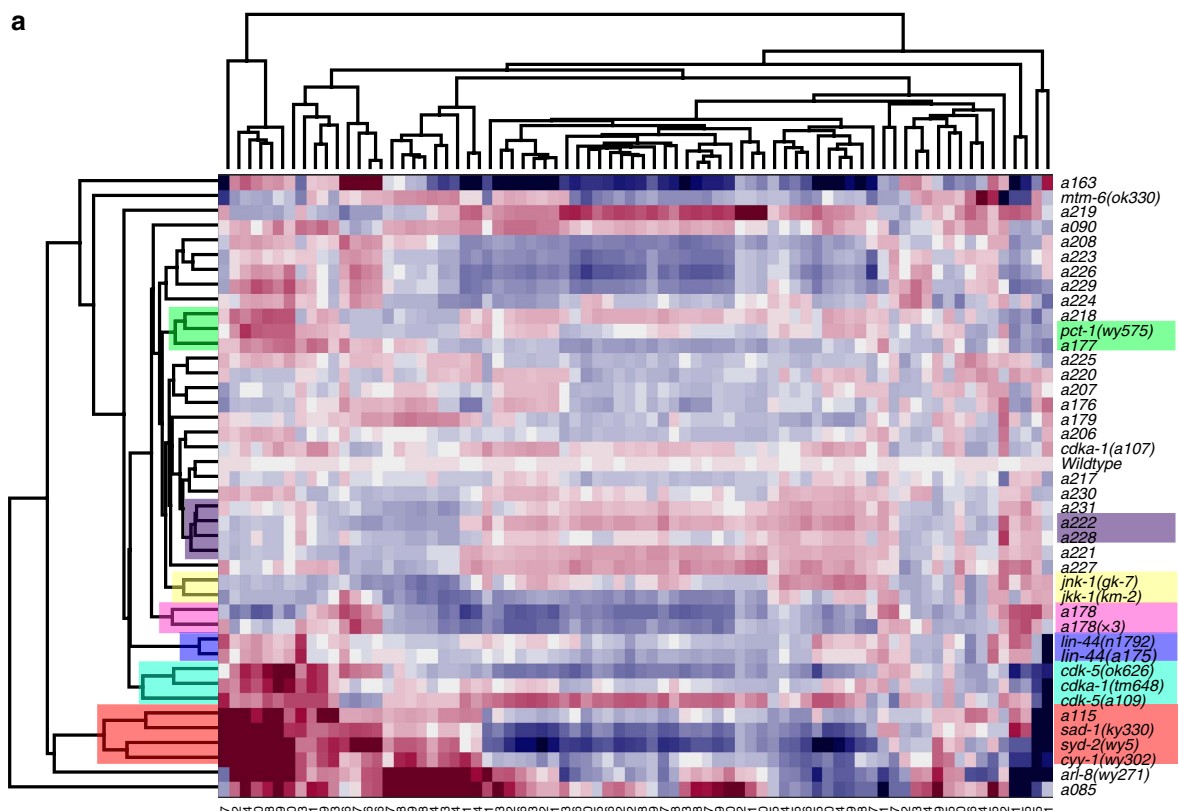

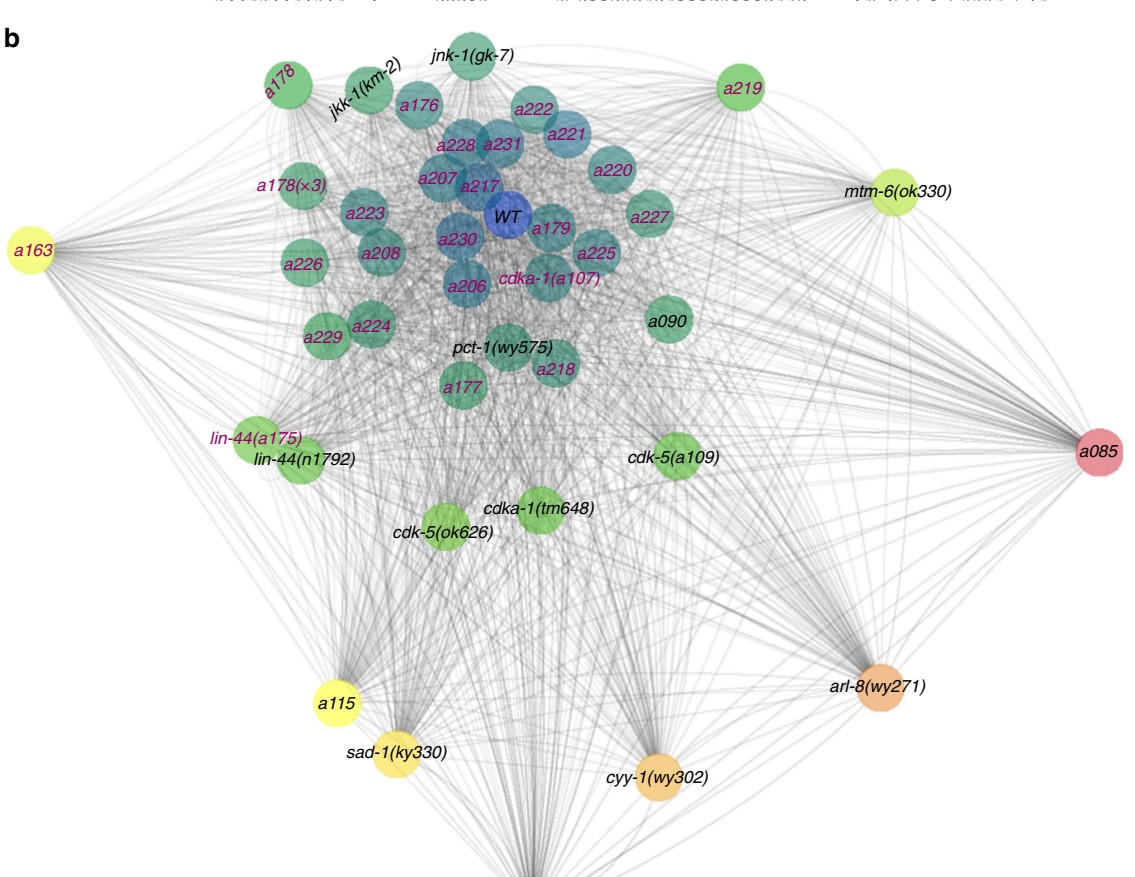

dependent upon JKK-1 activation[68], and were found in suppressor screens of *arl-8*, not a direct visual screen[58]. This example demonstrates that even in instances when visual inspection fails to detect differences, deep phenotyping can extract quantitative information that leads to identification of new alleles and possible functions. Another striking result of phenotypic clustering is that mutants with known dendritic mislocalization of synaptic puncta (for example, *cdka-1* and *cdk-5*) are also clustered together, even though the analysis is performed solely on the axonal synaptic domain. *cdka-1* and *cdk-5* are not only similar in their dendritic mislocalization, but are also known to work in the same pathway (*cdka-1* is an activator of *cdk-5* (refs 47,54)). Both new allele *a177* and *pct-1* exhibit dendritic mislocalization of synaptic material and are also clustered together. In addition, *a222* and *a228*, which are clustered, also exhibit dendritic mislocalization. The SWLR models for these mutants provide information regarding the phenotypic changes observed by these subtle mutants. The axonal phenotypes suggest that the dendritic mislocalization known for *cdka-1* and *cdk-5* is accompanied by a reduction and redistribution of synaptic material in the axonal domain. By further examining the SWLR models for these mutants, we find, for instance, the most relevant features for *cdka-1* and *cdk-5* (Supplementary Fig. 6): both include a reduced number of puncta and a slight shift in the location of the most posterior puncta, and reduced puncta intensity in the posterior portion of the synaptic domain. This further corroborates the idea that the synaptic material distribution on the dendritic and axonal regions are commonly regulated.

## Discussion

In this work we have applied an integrated approach to perform quantitative and unbiased phenotyping of synaptic morphology in *C. elegans* to search for alleles with very subtle but significant differences, extremely difficult to perceive by human vision. For genetic model systems, visual screens for subtle phenotypes is generally challenging because it is experimentally difficult to use many markers and to phenotype a large number of animals in isogenic populations. The approach we present here can address these challenges. Our high-throughput live imaging set-up enables fast acquisition of images leading to large sample numbers. Furthermore, coupling computer vision tools with the extraction of complex non-intuitive 'meta-features', a thorough quantitative reconstruction of phenotypic profiles is built, in addition to the improved speed and reduced human bias. These high-content profiles include far more information than what is perceptible to human vision, particularly those statistical in nature. We showed that this multidimensional phenotypic profiling overcame the confounding effects of phenotypic heterogeneity exhibited by isogenic populations, and enabled the identification and characterization of subtle mutants. Finding significant and relevant changes in the subtle mutants was enabled by stepwise feature selection applied to build logistic regression models and the pairwise population comparisons against wild type. We discovered extremely subtle mutants that

also display striking behavioural phenotypes, such as *a178*, impossible to identify by visual inspection, which illustrated the power of this approach to explore potentially previously unavailable phenospace.

Conventionally, mutants are traditionally considered interesting if they give a large enough phenotypic alteration, usually obvious to the eye; those alleles that have small changes, particularly where the heterogeneity of the samples drowns the difference, would be considered false-positives. The method presented extends the boundary of accessible phenospace to identify mutants, particularly those inaccessible via conventional manual scoring. This becomes more important as the work using genetic model systems move into problems relevant to human psychiatric diseases, for instance, where the morphological changes in the nervous system are extremely subtle[11,12]. Our analysis shows that detailed and precise phenotypic profiling is particularly effective at detecting differences from content-rich images, imperceptible to human vision. Quantifiable phenotypic changes, identified without prior knowledge or human vision-driven biases, should enable the identification of previously undiscovered morphological signatures that might aid in better understanding the relationships between genetic variations and phenotypic outcomes. This information can lead to hypotheses of altered biological functions within populations of animals where phenotypic differences appear extremely subtle. This type of analysis should be applicable to deep phenotypic profiling of fluorescent markers in the search for genes, metabolites or drugs important for various biological functions. We envision this approach to be invaluable in understanding the role of genes with no known function, and finding the missing relevant players in known pathways. Furthermore, this approach enables the identification of subtly varying phenotypes that can result from multigenic traits. Studies in model organisms, like *C. elegans*, can lead the way to develop analysis tools able to link phenotype to genotype in the complex but common scenarios of multigenicity and phenotypic pleiotropy, and reveal qualitatively uncovered phenotypic connections between genotypes.

## Methods

**C. elegans strains and culture.** Worms were cultured on nematode growth medium (NGM) plates seeded with OP50 *Escherichia coli* bacteria using standard methods[69]. Strains are listed in Supplementary Note 9. The wild-type strain is XA7810: N2, *wyIs85 (Pitr-1pB::gfp::rab-3)*. The following mutants were used *syd-2 (wy5)X*, *sad-1 (ky330)X*, *mtm-6 (ok330)III*, *jnk-1 (gk7)IV*, *jkk-1 (km2)X*, *cdka-1 (tm648)III*, *cyy-1 (wy302)*, *pct-1 (wy575)*, *cdk-5 (ok626)III*, *lin-44 (n1792)I*, *arl-8 (wy271)*, *sax-2 (ky216)III* and *sax-2 (ot10)III*. Mutants found in a previous screen[45] were also used: GT085, GT090, GT107, GT109 and GT115. Imaging of populations was performed on age-synchronized worms grown to the start of egg-laying age.

**Microfluidic device operation and imaging.** Microfluidic chips were made by polydimethylsiloxane (PDMS) replica molding from an SU-8 master mold fabricated by photolithography, as previously described[45]. During imaging or genetic screening, age-synchronized worms were suspended in M9 buffer with 0.01% Triton X-100. A sealed plastic vial containing the worm suspension connected to the inlet of the microfluidic chip and to the pressure source was used to inject worms into the chip by pressure-driven flow. Imaging was performed at $\times$ 40 magnification in a compound microscope using an oil objective (numerical aperture = 1.4) with a Hamamatsu Orca D2 camera for simultaneous imaging of the red and green channels.

**Figure 7 | Multidimensional profiles reveal mutant relationships and differences in the phenotypic spectrum.** (**a**) Hierarchical clustering suggests altered pathways. Highlights represent known genetic or phenotypic relationships. Mutants known to work in the same pathway, but with no visible phenotype, are accurately clustered together (*jkk-1* and *jnk-1*). Mutations that cause mislocalization to dendritic domains are also clustered together, from their axonal, previously undetected, phenotypes alone (*cdka-1*, *cdk-5* and *cdka-1*). Mutants with very similar phenotypes, and known to work in synaptic vesicle clustering (*syd-2* and *sad-1*) also cluster together. Complementation test confirmed *a175* is a new allele of *lin-44*. A list of all clustered genotypes in order appears in Supplementary Materials. (**b**) Network representing the average differences between populations computed from a weighted phenotypic profile. Subtle mutants isolated in this screen, imperceptible by visual inspection, are labelled in pink. Subtle mutants lie closest to wild type, while most previously identified mutants are the farthest from wild type.

**Off-chip components and automated operation.** Off-chip components include an in-house built pressure control box with a digital I/O card (Pacdrive, Ultimarc), solenoid valves (Cole Palmer) for off-chip on/off flow control and a custom in-house built cooling system consisting of a peristaltic pump for flow of the cooling liquid, a Peltier and an in-house built heat exchanger. Actuation of the solenoid valves, the pressure box, the microscope stage and the camera for image acquisition was controlled by a Graphical User Interface (GUI) developed in MATLAB (Mathworks). Automated operation during imaging or screening was performed with pre-built worm detection, imaging, phenotyping and sorting functions called from the GUI, as well as with error-handling functions.

**Genetic screens and whole-population profiling.** Genetic screens were performed by imaging an age-synchronized F2 population as previously described[45] with standard techniques[70]. Briefly, mutagenesis was performed based on standard protocol using 50 mM EMS on L4 animals. F1s from ~20 P0s were pooled per plate; five to six plates were prepared. F2 embryos hatched overnight in M9 to synchronize and subsequently cultured on NGM plates with OP50. F2 young adults were imaged and mutants were obtained from the screens and picked to an individual plate for self-propagation. A total of ~4,100 F2 animals were phenotyped from four separate mutagenesis experiments.

**Behavioural assays.** Crawling was assessed by counting the number of ventral bends that each animal performed in a 1-min period. Wild type and a178 were assessed on the same seeded NGM plate. $N = 20$. Aldicarb sensitivity assays were performed as described elsewhere[60], with a 1-mM aldicarb concentration in NGM plates. $N = 20$. Swimming assays were performed by transferring single animals at the start of egg-laying to wells fabricated on agarose pads filled with water. Videos were acquired of the swimming animals (Supplementary Videos 1 and 2).

**Statistics and animal models.** Imaging sample sizes were chosen to ensure adequate power to detect changes in the variables contained in the phenotypic profiles (Supplementary Note 11). Images excluded from analysis were those that have fewer than five puncta, since this prevents computing certain features; moreover, animals with fewer than five puncta are likely very small, or have defects in reporter expression. Sample sizes are larger than 50 for all data sets, except for imaging repeats of mutant populations (a176-R2) or outcrossed mutant populations (a178(x3)). Sample sizes account for all analysed images. Details of sample sizes are reported in Supplementary Information. Sample sizes for behavioural experiments were 20 for on-plate assays, and 30 for swimming assays (per genotype). No sample randomization was performed in this work. Analysis of imaging data was performed in an automated manner, without user input, and was thus blind. Behavioural assays were also performed blind.

**Data availability.** Software and code for high-throughput imaging, phenotypic profiling and SWLR model construction are publicly available at github at: https://github.com/asanmiguel/SynapsePhenotyping, under an MIT licence. The raw data set of phenotypic profiling is also included at the github repository. Raw images are available upon request.

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

## Acknowledgements

We thank Dr Brani Vidakovic for assistance with data analysis and Dr Maria Gallegos for sharing strains. This work was supported by NIH (R01GM088333 to H.L. and K99AG046911 to A.S.M.). K.S. is an HHMI investigator. Some strains were provided by the CGC, which is funded by NIH Office of Research Infrastructure Programs (P40 OD010440).

## Author contributions

A.S.-M. and H.L. conceived the experiments and wrote the manuscript. A.S.-M. and M.M.C. developed the experimental platform and computer vision algorithm. A.S.-M. performed imaging experiments, genetic screens and data analysis. P.T.K. and K.S. developed strains. P.T.K., Y.Z., P.T.M. and K.S. performed genotyping analyses.

## Additional information

**Competing financial interests:** The authors declare no competing financial interests.

