## [Peer Review File · Nature Communications]

Reviewers' comments:

Reviewer #1 (Remarks to the Author):

"Deep Phenotyping Unveils Hidden Traits and Genetic Relations in Subtle Mutants" by San-Miguel et al. describes an updated and extended version of their previous synapse phenotyping method (Nature Methods, 2012) and its application to a forward genetic screen. Improved methods for quantitative high-throughput phenotyping are important and this paper presents an interesting method and some new mutants.

Specific comments:

0) This is simply a question of style which the authors should feel free to ignore, but I felt too many sentences were highlighted by leading adverbs (interestingly is used 5 times, notably 3 times, remarkably twice, and strikingly once).

1) In the abstract, the authors state the limited number of fluorescent markers that can be incorporated in vivo is a bottleneck for phenotyping. This may be true, and they show that it is still possible to get useful information from a single marker, but this limitation is not really addressed in the current paper. For example, their method would also benefit from more independent markers to derive more informative phenotypes.

2) "the bottleneck for discovery of cellular functions now is phenotypical analyses" This is widely claimed, but would be clearer if specific examples (or perhaps a review with examples) were cited.

3) "have subtle phenotypes not necessarily accessible to eye." Again, this statement should be cited.

4) "particularly on whole animal morphology" This statement is misleading. Ref. 17 describes a wide variety of phenotypes, ref. 18 also includes high throughput assessment of fluorescence localisation, and ref. 19 measures high-dimensional phenotypes most of which are based on behavioural dynamics, not morphology.

5) "for the first time in multicellular systems" This statement of priority is both untrue and unnecessary for the paper.

6) "We developed a data analysis pipeline for unsupervised image annotation" This should be clarified. As the authors know, unsupervised has a specific meaning in machine learning and several steps of the analysis are in fact supervised learning methods.

7) "conventional notion of the stereotyped development". My understanding of the conventional notion was that the cell lineage is stereotyped, but not necessarily sub cellular features or cell body locations, which are known to vary between wild type animals.

8) "Interestingly, this also suggests that the screens that identified the mutants based on a single trait must have relied on favorable sampling of these populations (either by having screened more than one animal from these genotypes, or by having stochastically sampled a phenotypically severe individual); in other words, many alleles could have been missed in these screens even in screens for severe phenotypical changes." Could the authors provide citations to the kind of studies they are referring to? I'm asking because most manual phenotype studies are not based on single quantitative features, but on a complex human-defined phenotype, which is a very different comparison. For example, it's common for people to put animals into categories of phenotypic severity based on visual assessment. Such visual assessments can in fact integrate many features.

9) "Specifically, animals were positively sorted if any of the phenotypical features exceeded a pre-set threshold" Why not use an outlier detection method based on wild type data as in the Nature Methods 2012 paper? In the supplement, the authors state that this was done to avoid overfitting, but the manual choice also risks introducing bias. One of the main strengths of the approach in the current manuscript is the combination of multiple features to define phenotypes, as emphasised in the text and in figure 1, so I would expect that a multidimensional outlier detection method would work better.

10) "Only non-significant differences displayed" in Fig. 4 caption. Why not show significant differences as well?

11) "Best Feature (BF) Models (which mimics best-case-scenario manual detection)," As discussed above, manual human defined features will often be multi-dimensional. I agree that in many cases they will also be less sensitive than quantitative methods, but that's not demonstrated by this comparison.

12) "SWLR in general does not have to sacrifice sensitivity for specificity, or vice versa" I would remove 'in general' from this statement. It may be true compared to a single feature classifier on this data, but in general, there is still a sensitivity/specificity tradeoff inherent in the method.

13) "Another striking result of phenotypic clustering is that mutants with known dendritic mislocalization of synaptic puncta (e.g. *cdka-1* and *cdk-5*) are also clustered together, even though the analysis is performed solely on the axonal synaptic domain" It might be worthwhile to elaborate on this point. Which features are important in this case? Do they give a hint about the nature of the pleiotropic effect?

14) "The method presented here does not change the notion of a mutant" This should be rephrased. I think I understand what the authors are trying to say here, but the notion of a mutant is clear from the genotype and so of course will not change with a phenotyping method.

15) "Analysis of imaging data was performed in an unsupervised fashion". See point 6.

16) Under acknowledgements, the CGC's preferred text is "Some strains were provided by the CGC, which is funded by NIH Office of Research Infrastructure Programs (P40 OD010440)."

17) In the supplement: "false positive rate is negligible" The actual false positive rate (including how many cases were checked) should be included. Later: "the occurrence of such false positives, however, was extremely rare." How rare?

18) Supplement, second screen: "62 [mutants] were viable". Do you mean that 62 were kept after subsequent screening or that they were viable in the sense of producing viable offspring?

19) The multiple screening rounds described in the supplement rely on a large number of ad hoc analysis steps. Again, this is not exactly 'unbiased' analysis. There's not necessarily anything wrong with this, but without follow-up analysis it's impossible to know how many new alleles may have been discarded in the various filtering steps, nor how this would affect the clustering observed in the main figures if these other mutants had been kept.

20) figures:

-some extra dots around Fig. 5c

-font in 5e probably too small

-fonts in 6 also too small. I like the idea of the schematic representation in 6.

Reviewer #2 (Remarks to the Author):

Overall, the paper takes a nice approach of clustering genetic perturbations (alleles in *C. elegans*) based on measurement of multiple phenotypic features (using single-channel images of labeled synapses). As far as I can tell, the paper uses a previously-developed image processing pipeline (if I'm mistaken, the main text should include explanation of how the system differs), so what is new here is the application to a particular data set, and clustering of alleles based on phenotypes. It should be noted that I did not have time to closely review the extensive supplementary information so relied primarily on the main paper.

The paper was a bit difficult to assess given the inflated language implying that the basic approach is unprecedented (e.g., "We demonstrate, for the first time in multicellular systems, an approach to identify genetic alterations that give rise to subtle and varying phenotypical changes, unintuitive to and unobservable by human, but nonetheless relevant."). It may be true that this is the first time multicellular mutant samples have been grouped based on subtle phenotypic features but it is fundamentally the same approach as has been successfully applied to multicellular chemically-perturbed samples in the work from Randy Peterson's lab (zebrafish) and to genetically-perturbed fluorescently labeled cultured cells (Boutros lab and many others). As another example, the premise of this sentence is in my opinion untrue: "The lack of unbiased quantitative data extraction and analysis methods to identify subtle changes by weak alleles, (22, 23), means that the rich information encoded in fluorescence images of multicellular models has not been fully exploited." The references are more than 5 years old, and in fact, numerous subtle phenotypic changes have been quantified in multicellular models and quantitative data extraction and analysis methods indeed exist to do so; there are so many examples it is difficult to summarize - a google search for "Zebrafish image analysis" or "Worm image analysis" will attest. Furthermore, the introduction sets up a very general problem that the authors imply they have solved (subtle phenotyping in multicellular organisms), but their actual work is very specific, clustering mutant *C. elegans* alleles based on synapse phenotypes.

At first reading, the paper seems to present one biological discovery: that a178 has the SWIP phenotype. I'm not equipped to judge the impact of this finding. The remaining results in the paper refer to existing literature for validation, on an anecdotal basis. It is commonplace that evidence can be found to support most hypotheses, even randomly generated ones, in biology. Thus, it is difficult to assess the overall efficacy of the approach without either novel discoveries or quantitative validation against expected ground truth (i.e., which genes ought to show similarity to others, as specified prior to doing the experiment rather than after seeing the results). Given that so much of the paper relies on the validation of models on same-gene (vs. other-gene) samples, it is important to reassure the reader in the main text that holdout procedures were done properly; that is, that any images used to build a model come from a completely separate experiment from images used to test the model. I suspect this was not the case based on Supplementary Data 4 which indicates that each allele was tested in one and only one imaging round, with only a few exceptions. This makes all specificity/sensitivity conclusions of the paper suspect.

In several instances, the authors claim that certain metrics are invisible to humans and in some cases the claims are untrue (e.g., p.6 claims that statistical features such as variability are inherently outside considerations of human perception, and even more obviously untrue claims that taking into account multiple phenotypic features is also infeasible for humans; in fact humans are very good at

this). It is important to recognize that humans can indeed assess variability by eye, and can indeed observe multiple features simultaneously, especially in the description of the Best Feature method: it is described as a mimic for best-case-scenario manual detection but it is not. That analysis should be described as single-feature vs multi-feature, not computer vs human as is currently implied.

A major claim of the paper is that 'invisible' phenotypes are identified and a178 is presented as the main example of this. But to my eye, a178 seems quite visually distinct from WT; the puncta are clearly dimmer and more closely spaced (that is, there are more of them) and I detected this prior to reading the narrative. I can't assess this for the other two (a220 and a228 in Fig 3) because no WT images are shown side by side in the main text, and they are missing scale bars.

In several instances, the strength of the results is over-stated, if I understand them correctly. For example, a major claim of the paper relates to specificity. Two types of evidence are presented. The first is an overview of sensitivity/specificity (e.g. Fig 5a) which shows that the sensitivity and specificity are both less than 50% for the majority of alleles detected as subtle in this study; this is quite poor. I also suspect making comparison against BF (Single Best Feature) for assessing specificity is not sufficient, as the classification problem is inherently difficult in detecting both positives (sensitivity) and negatives (specificity) when only a single feature is used. So it is not surprising that use of BF results in lower both specificity and sensitivity. The second line of evidence relies on exploration of a178 ("For instance, a178 is exclusively identified by its own model and by the model for a178(x3), and vice versa (Fig. 5c,d).") Yet Fig 5c shows significant cross-talk with probabilities >0.5 for many other alleles, such as a163 and ky330; this is similarly shown in Fig 5b, which shows a163 as much higher than a175 even though a175 is described as 'barely' recognized by the model and a163 is ignored. Meanwhile, Fig 4c shows that the model built on a178(x0) yields only ~ 0.65 phenotypic probability for a178(x3), when a perfectly-performing system would achieve 1.0. The other sax-2 alleles shown in Fig 4c are said to "phenocopy" but in some cases only reach ~ 0.4 . Meanwhile in figure 5c, levels >0.4 are routinely seen from unrelated alleles but dismissed by the authors who claim in 5d that models are specific only to a178. I found this very confusing, why <0.5 in some analyses was deemed completely insignificantly similar and in others completely phenocopying.

The main text should contain a statement about software/code availability and license; the authors do not seem to indicate that the code is publicly available.

Technical points:

In building the model using step-wise logistic regression :

- 1) the likelihood should be evaluated on a held-out data to prevent overfitting.
- 2) p-values should be adjusted as multiple successive hypothesis tests are performed.

Suggestions:

- the main text should mention how many features were measured in the initial statement "we performed multi-dimensional profiling of the synaptic punch in DA9" and some description of what types. This is important for understanding the overall method (and comparing to the later set of 76 metrics mentioned on p.5) and shouldn't be buried in supp data.
- p.4 "metrics that fully capture phenotypes" mistakenly implies it is possible to truly fully capture a phenotype using image-based metrics. Similarly, p. 5 "we thus conducted comprehensive phenotypic profiling" - the use of "comprehensive" is inaccurate.
- replace phenotypical with phenotypic
- p. 5 "cannot be identified from phenotyping a single animal" refers to reference 26 but it is not obvious how that reference supports the statement; it needs some elaboration (did that paper neglected to identify that mutant? did the paper do a comprehensive analysis of how many animals

need to be examined to identify the phenotype?)

- p. 9 the main text needs more description of how the artificial images are constructed. In fact, I believe the term "schematic" would be more appropriate than "image" because the depictions are very simplified views of the phenotypes and are not at all similar to synthetic images (based on a full model of cell/organism appearance) that are sometimes produced in this field. The term "can be represented in a visual manner" implies that some features cannot be so represented, but again it is indeed possible to construct synthetic images from raw measurements (see the work of the Bob Murphy lab, for example). So "can easily be represented" is more accurate.
- p. 9 the description of Fig S9 is not informative; it doesn't explain what sort of representation is presented. The supplement does not have a legend for a Fig S9 (my guess is the legend for S8 is the one meant, so there is a supp figure missing).
- no reference to figures/data is provided for the statement "We subsequently tested this hypothesis and indeed found that a175 is an allele of lin-44 by complementation tests."
- p. 11 "is extremely subtle" > "are extremely subtle"
- Fig 1 b and c are not very explanatory; perhaps they need more text to explain (e.g. replace human head with "human-scored phenotype"), or should be omitted.
- Fig 2c shows 'relevant features' designated by numbers; the names of the features should be in the legend or on the plot.
- Fig 5c legend needs to explain what the size of each dot represents and explain the gray/green boxes at the lower left.
- Fig 5e legend needs to explain what the colors represent
- Fig 5a legend needs to explain how samples are ordered, left to right

REVIEWERS' COMMENTS:

Reviewer #1 (Remarks to the Author):

The authors have addressed my main concerns.

We thank the reviewers for their careful read of the manuscript and their valuable comments and suggestions. We are happy to hear that the reviewers find the manuscript “interesting” and “a nice approach”. We have carefully considered the comments and responded with changes in the revised manuscript. Below is a point-by-point response to each reviewer. The original comments are in black, not bolded, and our responses are in blue and bolded. Whenever possible, we also refer to the specific changes made in the manuscript.

Reviewers' comments:

Reviewer #1 (Remarks to the Author):

"Deep Phenotyping Unveils Hidden Traits and Genetic Relations in Subtle Mutants" by San-Miguel et al. describes an updated and extended version of their previous synapse phenotyping method (Nature Methods, 2012) and its application to a forward genetic screen. Improved methods for quantitative high-throughput phenotyping are important and this paper presents an interesting method and some new mutants.

Specific comments:

0) This is simply a question of style which the authors should feel free to ignore, but I felt too many sentences were highlighted by leading adverbs (interestingly is used 5 times, notably 3 times, remarkably twice, and strikingly once).

We appreciate the reviewer’s suggestions. The number of leading adverbs has been reduced throughout the text, e.g.

- **Page 5, 1st paragraph: “interestingly” replaced with “therefore”**
- **Page 6, 1st paragraph, “interestingly” removed from sentence “The most significant feature selected for *a178*”**
- **Page 6, 2nd paragraph, “remarkably” replaced with “nonetheless”**
- **Page 7, 2nd paragraph, “notably” removed from sentence “These animals display a distinct...”**
- **Page 7, 2nd paragraph, “notably” replaced with “significantly”**
- **Page 8, 1st paragraph, “Interestingly” replaced with “as expected”**

1) In the abstract, the authors state the limited number of fluorescent markers that can be incorporated in vivo is a bottleneck for phenotyping. This may be true, and they show that it is still possible to get useful information from a single marker, but this limitation is not really addressed in the current paper. For example, their method would also benefit from more independent markers to derive more informative phenotypes.

We agree with the reviewer’s comments, particularly that everyone would benefit from being able to use more markers simultaneously. Our point in the manuscript, perhaps stated more awkwardly than could have been, is that even in the limiting case with a single marker, it is still possible to extract large amounts of information, and this is information that was not formerly exploited fully. From this rich information, we were able to quantify previously unidentified differences among genotypes. The main goal of this exercise was to

show-case high-content quantitative phenotyping as a very useful tool. Again we agree that including more markers would be even better at phenotyping synaptic sites, and we hope to expand our capabilities to a larger number of markers in the future.

We have changed the text in the abstract to better reflect the point we are trying to make, “and extracting comprehensive information from the limited number of reporters that can be simultaneously incorporated in live animals”

2) "the bottleneck for discovery of cellular functions now is phenotypical analyses" This is widely claimed, but would be clearer if specific examples (or perhaps a review with examples) were cited.

The following citations have been added.

- 1. Houle, D., Govindaraju, D.R. & Omholt, S. Phenomics: the next challenge. *Nat Rev Genet* 11, 855-866 (2010).**
- 2. Furbank, R.T. & Tester, M. Phenomics – technologies to relieve the phenotyping bottleneck. *Trends in Plant Science* 16, 635-644.**
- 3. Deans, A.R. et al. Finding Our Way through Phenotypes. *PLoS Biol* 13, e1002033 (2015).**
- 4. Granier, C. & Vile, D. Phenotyping and beyond: modelling the relationships between traits. *Current Opinion in Plant Biology* 18, 96-102 (2014).**
- 5. Boutros, M., Heigwer, F. & Laufer, C. Microscopy-Based High-Content Screening. *Cell* 163, 1314-1325 (2015).**

3) "have subtle phenotypes not necessarily accessible to eye." Again, this statement should be cited.

A few citations have been added.

- 1. Houle, D., Govindaraju, D.R. & Omholt, S. Phenomics: the next challenge. *Nat Rev Genet* 11, 855-866 (2010).**
- 11. Ebert, D.H. & Greenberg, M.E. Activity-dependent neuronal signalling and autism spectrum disorder. *Nature* 493, 327-337 (2013).**
- 12. Fromer, M. et al. De novo mutations in schizophrenia implicate synaptic networks. *Nature* 506, 179-184 (2014).**

4) "particularly on whole animal morphology" This statement is misleading. Ref. 17 describes a wide variety of phenotypes, ref. 18 also includes high throughput assessment of fluorescence localisation, and ref. 19 measures high-dimensional phenotypes most of which are based on behavioural dynamics, not morphology.

We thank the reviewer for pointing out the mis-statement. Our intention was to emphasize previous methods for having focused on features larger than sub-cellular resolution ones. We have re-phrased this statement to: “In *C. elegans*, high-throughput live phenotyping

has mostly focused on drastic changes exhibited on gross features (for example, whole animal or tissue-level changes)".

5) "for the first time in multicellular systems" This statement of priority is both untrue and unnecessary for the paper.

This sentence has been removed.

6) "We developed a data analysis pipeline for unsupervised image annotation" This should be clarified. As the authors know, unsupervised has a specific meaning in machine learning and several steps of the analysis are in fact supervised learning methods.

To avoid confusion, we have changed “unsupervised” to “automated”. As described in the supplemental information, image annotation is based on machine learning, where the training is in fact supervised.

7) "conventional notion of the stereotyped development". My understanding of the conventional notion was that the cell lineage is stereotyped, but not necessarily sub cellular features or cell body locations, which are known to vary between wild type animals.

The conventional notion in regards to the location and even number of synaptic connections is also stereotyped (White et al; Durbin; Bargmann; Hall and Russell). DA9 has been a great example of stereotyped synaptic connectivity (Varshne et al; Margeta et al). Sub-cellular features, such as presynaptic sites in DA9, as well as the location of all neurons in the animal’s nervous system is also stereotypical. Citations have been added in the text.

References:

- White JG, Southgate E, Thomson JN, Brenner S (1986) The structure of the nervous system of the nematode *Caenorhabditis elegans*. *Phil Trans R Soc Lond B* 314:1–340. doi:10.1098/rstb.1986.0056.
- Durbin RM (1987) Studies on the development and organisation of the nervous system of *Caenorhabditis elegans*. Ph.D. thesis, University of Cambridge.
- Bargmann CI (1993) Genetic and cellular analysis of behavior in *C. elegans*. *Annu Rev Neurosci* 16:47–71. doi:10.1146/annurev.ne.16.030193.000403.
- Hall DH, Russell RL (1991) The posterior nervous system of the nematode *Caenorhabditis elegans*. Serial reconstruction of identified neurons and complete pattern of synaptic interactions. *J Neurosci* 11: 1–22.
- Varshne LT, Chen BL, Paniagua E, Hall DH, Chklovskii DB (2011) Structural Properties of the *Caenorhabditis elegans* Neuronal Network *PLOS Comp Bio*, 7: e1001066
- Margeta MA, Shen K, Grill B (2008) Building a synapse: lessons on synaptic specificity and presynaptic assembly from the nematode *C. elegans* *Curr Opin Neurobiol.* 18: 69-76

8) "Interestingly, this also suggests that the screens that identified the mutants based on a single

trait must have relied on favorable sampling of these populations (either by having screened more than one animal from these genotypes, or by having stochastically sampled a phenotypically severe individual); in other words, many alleles could have been missed in these screens even in screens for severe phenotypical changes." Could the authors provide citations to the kind of studies they are referring to? I'm asking because most manual phenotype studies are not based on single quantitative features, but on a complex human-defined phenotype, which is a very different comparison. For example, it's common for people to put animals into categories of phenotypic severity based on visual assessment. Such visual assessments can in fact integrate many features.

We agree with the reviewer that indeed human could be looking for complex phenotypes, combining multiple quantitative features. Our intention was to communicate that traditional screens rely on identifying mutants that exhibit (the most) drastic changes among the mutagenized population. The phenotypes are dramatic, regardless of whether the phenotypes come from one or several quantitative features. We have also added a couple of citations for example screens that focus on identification of synaptic puncta mislocalization.

- **Klassen, M.P. et al. An Arf-like Small G Protein, ARL-8, Promotes the Axonal Transport of Presynaptic Cargoes by Suppressing Vesicle Aggregation. *Neuron* 66, 710-723 (2010).**
- **Tu, Haijun et al. *C. elegans* Punctin Clusters GABA_A Receptors via Neuroligin Binding and UNC-40/DCC Recruitment, *Neuron*, 86, 1407 – 1419 (2015)**

9) "Specifically, animals were positively sorted if any of the phenotypical features exceeded a pre-set threshold" Why not use an outlier detection method based on wild type data as in the Nature Methods 2012 paper? In the supplement, the authors state that this was done to avoid overfitting, but the manual choice also risks introducing bias. One of the main strengths of the approach in the current manuscript is the combination of multiple features to define phenotypes, as emphasised in the text and in figure 1, so I would expect that a multidimensional outlier detection method would work better.

We agree that an outlier method would be a better unbiased approach to identify mutants from a population. However, a randomly mutagenized population typically displays a significantly larger degree of variability than a wildtype population, since mutations are introduced that can affect the growth rate, the animal size, their feeding patterns, etc, which all affect the phenotypical output to a certain degree. To avoid screening these effects inadvertently by the outlier method, we first aimed at identifying mutants that displayed subtle changes in a few targeted characteristics, rather than taking an integrative multiparametric approach; practically the latter approach would have resulted in large numbers of false positives (data not shown). Given the practical difficulty involved in re-scoring large populations, we tried to reduce the rate of false positives with this approach. Our rationale was that animals that displayed subtle changes in synaptic puncta size and intensity etc. would be good candidates for subtle synaptic patterning mutations.

Nonetheless, the reviewer is correct in suggesting an outlier method as a less biased approach for screening; it is just that we chose to do the initial screens with the selected feature set for practical purpose. Our methodology here essentially allows us to expand the type of screens we can do in the future. For example, we could apply specific SWLR models to identify mutations that either phenocopy some of our identified mutants, or rescue the phenotypes of our isolated mutants, in which case the outlier method would be a good approach.

We have clarified the supplemental text with the rationale of our initial screens.

10) "Only non-significant differences displayed" in Fig. 4 caption. Why not show significant differences as well?

All comparisons are significant except those shown; in other words, WT is different from *sax-2* alleles including *a178*, and the *sax-2* allele *ky216* is different from the other alleles. We display only NS differences to emphasize that *sax-2(ot10)* phenocopies *a178*. We have added to the legend that all other comparisons are $P < 1e-6$

11) "Best Feature (BF) Models (which mimics best-case-scenario manual detection)," As discussed above, manual human defined features will often be multi-dimensional. I agree that in many cases they will also be less sensitive than quantitative methods, but that's not demonstrated by this comparison.

We agree with this comment, and have changed our language to show that BF models are essentially the results if only a single quantitative feature was used to characterize the mutants. The text now reads as follows:

“We apply the BF model as a best-case scenario when using single metrics to characterize the mutants. In reality, and without a prior feature selection algorithm (like the one performed here through SWLR), the results of the BF model are an optimistic estimate of the results when using a single-parameter approach.”

12) "SWLR in general does not have to sacrifice sensitivity for specificity, or vice versa" I would remove 'in general' from this statement. It may be true compared to a single feature classifier on this data, but in general, there is still a sensitivity/specificity tradeoff inherent in the method.

Changes made.

13) "Another striking result of phenotypic clustering is that mutants with known dendritic mislocalization of synaptic puncta (e.g. *cdka-1* and *cdk-5*) are also clustered together, even though the analysis is performed solely on the axonal synaptic domain" It might be worthwhile to elaborate on this point. Which features are important in this case? Do they give a hint about the nature of the pleiotropic effect?

We appreciate this comment from the reviewer. We indeed agree that this point should have been elaborated. Studying the most relevant features for these two mutants, we do

seem to find important changes in a few features. We have added some discussion regarding this point:

“The SWLR models for these mutants provide information regarding the phenotypic changes observed by these subtle mutants. The axonal phenotypes suggest that the dendritic mislocalization known for *cdka-1* and *cdk-5* is accompanied by a reduction and redistribution of synaptic material in the axonal domain. By further examining the SWLR models for these mutants, we find, for instance, the most relevant features for *cdka-1* and *cdk-5* (Fig. S6) both include a reduced number of puncta, and a slight shift in the location of the most posterior puncta, and reduced puncta intensity in the posterior portion of the synaptic domain. This further corroborates the idea that the synaptic material distribution on the dendritic and axonal regions are commonly regulated.”

14) "The method presented here does not change the notion of a mutant" This should be rephrased. I think I understand what the authors are trying to say here, but the notion of a mutant is clear from the genotype and so of course will not change with a phenotyping method.

We have rephrased as follows:

“The method presented extends the boundary of accessible phenospace to identify mutants, particularly those inaccessible via conventional manual scoring”.

15) "Analysis of imaging data was performed in an unsupervised fashion". See point 6.

We have changed both instances to “automated”.

16) Under acknowledgements, the CGC's preferred text is "Some strains were provided by the CGC, which is funded by NIH Office of Research Infrastructure Programs (P40 OD010440)."

Changes have been made.

17) In the supplement: "false positive rate is negligible" The actual false positive rate (including how many cases were checked) should be included. Later: "the occurrence of such false positives, however, was extremely rare." How rare?

We have included additional details about SVM model performance for synapse segmentation, the false positive rate for the training set is 0.85%. We have added this to the supplemental text

18) Supplement, second screen: "62 [mutants] were viable". Do you mean that 62 were kept after subsequent screening or that they were viable in the sense of producing viable offspring?

They were viable, meaning producing viable offspring; this is clarified in the supplemental text.

19) The multiple screening rounds described in the supplement rely on a large number of ad hoc analysis steps. Again, this is not exactly 'unbiased' analysis. There's not necessarily anything

wrong with this, but without follow-up analysis it's impossible to know how many new alleles may have been discarded in the various filtering steps, nor how this would affect the clustering observed in the main figures if these other mutants had been kept.

We certainly agree with the reviewer that the first step in the screening process is not “unbiased”; that is, we may have missed some interesting mutants, and we may have biased the selection towards more robust phenotypes. This is again limited by the practicality of the large screen. Given the level of complexity for phenotyping such large numbers of populations, we had to perform a first-pass filtering process to enrich for putative mutants with higher likelihood of carrying a phenotype (of interest).

Our second step for re-screening the putative mutants is unbiased, in the sense that we score the many features “blindly” and let the algorithm pick what might be relevant and which of the mutants were actually robustly different from wildtype (or from each other). The methodology is mainly relying on the second step. The first step is a “necessary evil”. In an ideal world where there is no limitation on time and resources, one would want to generate an isogenic population from each individual event from the mutagenesis and analyze them. We certainly did not mean that the entire screen presented here is a catch-all screen, but only that the approach we take in the second step (based on whole population phenotyping and SWLR) is unbiased and could pick up information missed otherwise.

It is also important to mention that when the putative mutants (from the screen in step one) were analyzed via multiparametric phenotyping, size differences (resulting from background mutations during mutagenesis) was rarely the most important feature. In other words, the algorithm is robust against this type of variations/noise and “focuses” instead on features that would be biologically sensible.

In the clustering step, all features were used, i.e. there is no preselection; adding more genotypes into the process should not affect how the existing mutants relate to one another – it should only add the new genotypes to existing clusters.

We understand that this may be a confusing point. We have modified the supplemental text to try to clarify the points stated here.

20) figures:

-some extra dots around Fig. 5c

Figure 5c has been re-plotted as Population tested vs model (whereas before it was Model vs. Population Tested). The colored dots shown are only those with a $p > 0.5$, all others are plotted in gray.

-font in 5e probably too small

Because of space limitations we only use a larger font for those genotypes where relationships were confirmed. We would be happy to include a larger version of this figure as a supplemental figure and remove the smaller font-sized text here, if that is helpful.

-fonts in 6 also too small. I like the idea of the schematic representation in 6.

We have increased the fonts of parts a and c, due to space limitations this is not possible for part b. However, we have included the ordered list for hierarchical clustering in supplemental material.

RESPONSES TO REVIEWER # 2.

Reviewer #2 (Remarks to the Author):

Overall, the paper takes a nice approach of clustering genetic perturbations (alleles in *C. elegans*) based on measurement of multiple phenotypic features (using single-channel images of labeled synapses). As far as I can tell, the paper uses a previously-developed image processing pipeline (if I'm mistaken, the main text should include explanation of how the system differs), so what is new here is the application to a particular data set, and clustering of alleles based on phenotypes. It should be noted that I did not have time to closely review the extensive supplementary information so relied primarily on the main paper.

We thank the reviewer for pointing out this potential confusion to the reader. As stated in response to Reviewer 1, the experimental work in this paper is a two-step screen. Step 1 is similar to Crane et al (2012) as well as parts of the image analysis pipeline. Step 2 is the new development and the essence of this work, in which we use SWLR to analyze populations of isogenic animals to discern subtle mutants, which was not done in Crane et al. In other words, we would not have been able to find many of the mutants here using algorithms from Crane et al.

In terms of implementation, we developed new models for: 1) Image segmentation (based on the previously used support vector machines); and 2) Feature extraction. The current method expands the sets of metrics extracted from an image to a larger set inclusive of complex features. Furthermore, a significant difference is important to note: our previous work was focused on imaging animals in the lateral orientation, and included metrics mostly related to location of synaptic sites within the whole neuron. In this work, we have developed a microfluidic device that is able to rotate animals to a dorsal-down orientation. In this way, we image exclusively the axon of the animal. Our image analysis thus contains very few metrics related to location (mainly where within the axon synaptic puncta are located), while all the others are metrics related to size, intensity, homogeneity, etc, which can better capture the differences exhibited by subtle phenotypes.

In order to better explain the details of how our image processing pipeline is different, we have included additional details in the main text (Page 4). For readers interested in exact implementation, details about differences in the new high-throughput imaging system (including the new microfluidic device design), the new image segmentation models, and the feature extraction are also included in supplemental information.

The paper was a bit difficult to assess given the inflated language implying that the basic approach is unprecedented (e.g., "We demonstrate, for the first time in multicellular systems, an approach to identify genetic alterations that give rise to subtle and varying phenotypical changes, unintuitive to and unobservable by human, but nonetheless relevant."). It may be true that this is the first time multicellular mutant samples have been grouped based on subtle phenotypic features but it is fundamentally the same approach as has been successfully applied to multicellular chemically-perturbed samples in the work from Randy Peterson's lab (zebrafish) and to genetically-perturbed fluorescently labeled cultured cells (Boutros lab and many others).

We have revised the manuscript with several changes related to the novelty of the work, mainly to put into the specific context. For instance, the cited sentence now reads: “We demonstrate an approach to identify genetic alterations that give rise to subtle and varying phenotypic changes, unintuitive to and unobservable by human, but nonetheless relevant in the nematode *C. elegans*”.

We do believe though that the work done in zebrafish and cells, as cited below, are either low throughput and using human to screen, or higher throughput but with smaller set of features (even though in some instances large amounts of data, e.g. behavioral data, are acquired) to do the phenotyping.

A review paper by Rennekamp and Peterson highlighted the state of art in zebrafish screen:

- Rennekamp and Peterson, *Current Opinion in Chemical Biology* 2015, 24:58–70

Several specific refs cited by this review illustrate the general strategy, including the following:

- Rihel et al, *Science* 15 Jan 2010: Vol. 327, Issue 5963, pp. 348-351, DOI: 10.1126/science.1183090
- Das et al, *PLoS ONE* 5(4): e10004. doi:10.1371/journal.pone.0010004

Behavior screen review paper:

- Kokel et al, *Trends in Biotechnology*, Volume 30, Issue 8, August 2012, Pages 421–425

As another example, the premise of this sentence is in my opinion untrue: "The lack of unbiased quantitative data extraction and analysis methods to identify subtle changes by weak alleles, (22, 23), means that the rich information encoded in fluorescence images of multicellular models has not been fully exploited."

The references are more than 5 years old, and in fact, numerous subtle phenotypic changes have been quantified in multicellular models and quantitative data extraction and analysis methods indeed exist to do so; there are so many examples it is difficult to summarize - a google search for "Zebrafish image analysis" or "Worm image analysis" will attest.

We appreciate the sentiment of the review. We agree that phenotypic analysis is simply an approach to quantify genetic or chemically-induced changes, and has been done many times in many systems. The point we are making here, however, is that the majority of these phenotypic screens for multicellular organisms (zebrafish, worms, or *Drosophila*), have (1) not focused on high-resolution features (i.e., sub-cellular features within a single cell of a whole living organism), which is technically more challenging, or (2) used fixed samples (such as in *Drosophila* embryos). Nonetheless, we appreciate the reviewer's comments, and have changed the language and included additional references.

We would like to emphasize that cell culture might not be the most appropriate comparison; this is because the technical limitations of identifying changes within a single

cell of a large multicellular organism and those in cultured cells are of different order of magnitude.

Furthermore, the introduction sets up a very general problem that the authors imply they have solved (subtle phenotyping in multicellular organisms), but their actual work is very specific, clustering mutant *C. elegans* alleles based on synapse phenotypes.

We added an additional sentence in the introduction to focus the work towards synaptic patterning. We thought it was pertinent to give a broad introduction, since this method of phenotyping (SWLR and clustering on high dimensional morphological data) could be beneficial not only for synaptic patterning, but also for other biological processes that use fluorescent markers, colorimetric markers, behavioral measurements, etc – many of these problems are difficult to assess with qualitative inspection, or where quantification is a challenge.

At first reading, the paper seems to present one biological discovery: that *a178* has the SWIP phenotype. I'm not equipped to judge the impact of this finding.

We would like to clarify that the finding that *a178* has a SWIP phenotype is significant, and that the discovery is also that *sax-2* is now implicated in synaptogenesis, which is previously unknown. The SWIP phenotype is important because the morphological phenotype exhibited by *a178* is extremely subtle when compared with traditional synaptic patterning mutants. For this reason, without a functional defect (such as swimming-induced paralysis), it was difficult to determine whether the morphological phenotype observed is in reality a relevant mutation that affects synaptic function. The fact that the SWIP phenotype was identified after the mutant was isolated from phenotypic profiling of synaptic sites shows that subtle morphological phenotypes can in fact be informative for functional defects.

We have included more details in the text (Page 7) to emphasize the relevance of this finding.

The remaining results in the paper refer to existing literature for validation, on an anecdotal basis. It is commonplace that evidence can be found to support most hypotheses, even randomly generated ones, in biology. Thus, it is difficult to assess the overall efficacy of the approach without either novel discoveries or quantitative validation against expected ground truth (i.e., which genes ought to show similarity to others, as specified prior to doing the experiment rather than after seeing the results). Given that so much of the paper relies on the validation of models on same-gene (vs. other-gene) samples, it is important to reassure the reader in the main text that holdout procedures were done properly; that is, that any images used to build a model come from a completely separate experiment from images used to test the model. I suspect this was not the case based on Supplementary Data 4 which indicates that each allele was tested in one and only one imaging round, with only a few exceptions. This makes all specificity/sensitivity conclusions of the paper suspect.

Method validation:

We appreciate the reviewer's points about method validation. The main reason we included a large collection of known mutants was in fact to validate our approach. It is very important to emphasize that from the known mutant collection, a large portion of these were initially isolated because of large mislocalization defects, which are very easy to identify visually. We imaged these mutants without prior expectations (i.e. treating them as any other populations) and asked the algorithm to select the features automatically, and eventually clustered them accordingly. We are able to identify these mutants only from their axonal synaptic patterning, which has not previously been identified as defective. Thus, these known mutants are in fact the validations of our method.

Furthermore, the fact that we identify one of our mutants as exhibiting behavioral phenotypes (indicative of defective synaptic function) shows that the subtle mutants identified with this method have relevance on neuronal functions. Our method is effective since these morphological phenotypes would be extremely difficult to identify and isolate with traditional techniques, due to a combination of technical limitations of *C. elegans* culture, and difficulty in quantifying such subtle changes at a smaller scale. We show here that we were able to not only identify previously isolated animals, from a completely different characterization approach (i.e., axonal patterning), but also able to prove functional defects for one of our subtle mutants.

Imaging and holdout procedure:

We note here that we actually performed the imaging of mutants and always paired with wildtype control in each round as a normalizing metric (as explained in supplemental detail). This is because different rounds of imaging might exhibit noise inherent to experimental conditions, such as the batch-to-batch variations of agar source and bacterial culture used to grow the animals, the amount of food the animals consume during culture, the intensity of the fluorescent light source (which can vary during its lifetime), etc., all of which are common to all imaging experiments of this sort. This direct comparison between mutants and the wildtype population cultured and imaged using the same batch of reagents on the same day reduces the noise in the data. Models built to distinguish these populations actually perform better because they largely disregard the day-to-day variation. It is also important to note that the number of animals for each of the imaging rounds is large, as shown in the supplemental data.

To directly address the reviewer's concerns, we have, taking advantage of the large data set, tested our models not only in a hold-out procedure, but in a 5-fold cross-validation, which should better inform how the models behave. (This point is further explained in detail later in response to another comment to the reviewer.) Briefly, Figure 3b has been modified to plot AUC as a metric of model performance, rather than model fit. Each value represents the average AUC from the cross-validation rounds, along with error bars representing SEM. As can be seen from these plots, the models are able to achieve high-levels of cross-validation.

Regarding sensitivity and specificity

It is important to mention that the results presented in the initial submission of this manuscript were using metrics different from the usual definitions for sensitivity (true positive rate) or specificity (true negative rate). We had tested how much better each single model was at detecting the mutants it was meant to, as compared to all other mutants, and this had been plotted as specificity. For sensitivity, we had elected to use a very low false positive rate (which is where we would like to operate when running a genetic screen), and calculated the rate at which we picked up true mutants.

In order to prevent confusion, we have updated these results and applied the traditional specificity and sensitivity definitions. As can be seen in Figure 5a, when using traditional metrics (i.e., true positive rate and true negative rate using a threshold of 50% probability), most models have a sensitivity above 50% for SWLR, but not for BF, as expected. The same is true for specificity, although there is not much difference in performance for SWLR and BF in this scenario. Details have been added in the text to discuss these results as plotted in the revised figures. Furthermore, we added a supplemental figure (Figure S9) to show the accuracy of BF and SWLR models, where we confirm that SWLR results in higher accuracy, without sacrificing performance (as measured by the standard deviation of the 5-fold cross-validated accuracy).

In several instances, the authors claim that certain metrics are invisible to humans and in some cases the claims are untrue (e.g., p.6 claims that statistical features such as variability are inherently outside considerations of human perception, and even more obviously untrue claims that taking into account multiple phenotypic features is also infeasible for humans; in fact humans are very good at this). It is important to recognize that humans can indeed assess variability by eye, and can indeed observe multiple features simultaneously, especially in the description of the Best Feature method: it is described as a mimic for best-case-scenario manual detection but it is not. That analysis should be described as single-feature vs multi-feature, not computer vs human as is currently implied.

We understand where this reviewer may be coming from. We agree that human vision is a good at compounding information. However, we would like to take this opportunity to respond to a couple of specifics. First, it is well known that human vision is highly biased by the context – the information in the surrounding. Second, severe variability in strong phenotypes, such as allele *a085* (images in supplemental Figure 5a), could be easy to identify, although in a microscopy setting, to be exact in pin-pointing the variability being higher or lower than that of wildtype would probably still be very difficult. Other less obvious traits would be even harder in our opinion.

The reviewer would probably agree with us that computerized phenotyping would be pulling out information that is statistical in nature, at least different from (if not better than) human perception. The point we would like to make in the manuscript is that the multi-parametric quantitative analysis should be able to analyze phenotypes in an unbiased manner, and should incorporate complex information, which could push beyond human perception. We have re-examined the language we used in the text to try to clarify these points and present a balanced view in terms of the comparison to human perception.

A major claim of the paper is that 'invisible' phenotypes are identified and a178 is presented as the main example of this. But to my eye, a178 seems quite visually distinct from WT; the puncta are clearly dimmer and more closely spaced (that is, there are more of them) and I detected this prior to reading the narrative. I can't assess this for the other two (a220 and a228 in Fig 3) because no WT images are shown side by side in the main text, and they are missing scale bars.

We agree that after detailed study (and 3x back crosses to get rid of background mutations), allele *a178* might look different from wildtype by eye. However, it is important to emphasize that 1) isolating this allele, and 2) confirming its differences against wildtype without a doubt would not be possible without quantitative phenotyping. In fact, anecdotally, we had initially sent the mutant to be phenotyped by several people who also work with this particular marker, and all had called the phenotype “pretty much wildtype”. Traditional synaptic patterning phenotypes are drastically different, where synapses are created in completely different locations within a neuron. Proving that this is an allele worth studying further for traditional *C. elegans* neuroscientists without robust, reproducible quantitative information would have been difficult.

Nonetheless, we have taken into consideration the reviewer’s comments, and have changed the language to imply this is a subtle, but not invisible phenotype.

We have included a wildtype in Figure 3, for comparison, along with scale bars.

In several instances, the strength of the results is over-stated, if I understand them correctly. For example, a major claim of the paper relates to specificity. Two types of evidence are presented. The first is an overview of sensitivity/specificity (e.g. Fig 5a) which shows that the sensitivity and specificity are both less than 50% for the majority of alleles detected as subtle in this study; this is quite poor. I also suspect making comparison against BF (Single Best Feature) for assessing specificity is not sufficient, as the classification problem is inherently difficult in detecting both positives (sensitivity) and negatives (specificity) when only a single feature is used. So it is not surprising that use of BF results in lower both specificity and sensitivity.

Please see note above regarding sensitivity and specificity.

The second line of evidence relies on exploration of a178 ("For instance, a178 is exclusively identified by its own model and by the model for a178(x3), and vice versa (Fig. 5c,d).") Yet Fig 5c shows significant cross-talk with probabilities >0.5 for many other alleles, such as a163 and ky330; this is similarly shown in Fig 5b, which shows a163 as much higher than a175 even though a175 is described as 'barely' recognized by the model and a163 is ignored.

In order to clarify this point, figure 5c has been plotted as Population tested vs. Model (instead of Model vs. population tested as in the original). Although model *a178* seems to identify *a163* with a probability close to 0.5, the only other two sets identified by this model are *a178(x3)* and *a178*. This may mean that *a163* could be more closely related to or have a genetic interactions with *a178* than with any other genes/alleles tested (other than itself). The reviewer is correct that the equivalent information is shown in Figure 5b.

The plotted results in Figure 5b are the average population mutant phenotypic probability, obtained from the model for *a178*. There is no reason why allele *a175* should be identified by this model. *a175* is the least subtle allele isolated in this screen. The fact that it is not identified by model *a178* only implies that the features relevant for distinguishing allele *a178* against wildtype are not those features that distinguish *a175* from wildtype. As shown in Figure 3b, alleles *a163* and *a175* are the most easily identifiable, as measured by the goodness-of-fit and model performance metric AUC.

Meanwhile, Fig 4c shows that the model built on *a178(x0)* yields only ~0.65 phenotypic probability for *a178(x3)*, when a perfectly-performing system would achieve 1.0.

There are two important pieces to explain here. First, *a178(x3)* is backcrossed 3x with wildtype, so many mutations present in *a178(x0)* are not present in *a178(x3)*, which could cause small differences in phenotypes. Second, a “perfectly performing system” would not necessarily achieve an average probability of 1 (implying all mutants are scored at 1) -- due to inherent biological stochasticity, some of the animals might look more wildtype than others (and some wildtype don't necessarily look like an average wildtype). This touches on the issue of penetrance. We cannot assure that all animals must exhibit a mutant phenotype with a probability of 1. This is also the reason we perform the studies by comparing populations rather than single animals. Although not perfect, it is reassuring that *a178(x3)* is the highest scoring population after *a178(x0)* itself among a large number of genotypes.

The other *sax-2* alleles shown in Fig 4c are said to “phenocopy” but in some cases only reach ~0.4. Meanwhile in figure 5c, levels >0.4 are routinely seen from unrelated alleles but dismissed by the authors who claim in 5d that models are specific only to *a178*. I found this very confusing, why <0.5 in some analyses was deemed completely insignificantly similar and in others completely phenocopying.

We thank the reviewer for point out this potentially confusing issue for the general readership. The reviewer is correct in pointing out that *sax-2(ky216)* is an allele of *sax-2* (and in fact the only genotype) that does not phenocopy *a178* (by our criterion of $P < 0.5$). This is not inconceivable because different alleles may affect the gene functions differently, and the data suggest that *ky216* mutation had a different effect on synaptic morphology from *a178* and *ot10* do. In addition to whole genome sequencing and SNP mapping pointing to *sax-2*, we use this information to corroborate that *a178* is yet another allele of *sax-2*. We have clarified this in the text.

The main text should contain a statement about software/code availability and license; the authors do not seem to indicate that the code is publicly available.

The code for:

1. Graphical User Interface for High-throughput imaging
2. Image segmentation through SVM
3. Phenotypic profiling metric quantification
4. SWLR model fitting

is publicly available at github at: <https://github.com/asanmiguel/SynapsePhenotyping>, under an MIT license. The raw data set of phenotypic profiling is also included at the github repository. Raw images are available upon request.

This information has been added as a note to the article.

Technical points:

In building the model using step-wise logistic regression :

1) the likelihood should be evaluated on a held-out data to prevent overfitting.

We appreciate the thorough detail the reviewer has provided on this point. We agree that a hold-put approach or the more rigorous k-fold validation method is preferred to test the goodness-of-fit of the models, which we performed on data in Figure 3b. These values measure the area under the curve (AUC), for a receiving operating characteristic curve for each model. The plotted values are the mean AUC for a 5-fold cross-validation, and the error bars are the standard deviation obtained from the 5 tests. As can be seen from Figure 5b, the standard deviations are rather small, which implies that the models shouldn't be overfitting. We want to point out that in this 5-fold validation procedure, all the data has been used for testing and training independently.

2) p-values should be adjusted as multiple successive hypothesis tests are performed.

In the construction of SWLR models, p-values are used as a threshold for a variable's inclusion into or elimination from the model. These values are based on the deviance of the model with the new variable added or removed, as compared to the previous model, based on the Chi-squared statistic. Varying the p-value as the models are constructed would mean that variables are given different weight for entry or exit, depending on how far down in the step-wise logistic regression model fitting iteration these are included or excluded. It is common practice to use a constant value of $p=0.05$ for entry and $p=0.1$ for exit of variables (Hosmer, D. W., Jr., S. A. Lemeshow, and R. X. Sturdivant. 2013. Applied Logistic Regression. 3rd ed. Hoboken, NJ: Wiley.). Given that at each model construction, only each mutant population is compared against wildtype (not all mutants against each other), we do not apply a multiple-comparison adjustment to the p-values. In cases where multiple comparisons are being made (such as Figure 4c), we did apply a multiple-comparison correction (i.e., Bonferroni correction). We made sure the text is clear about these procedures and rationales.

Suggestions:

- the main text should mention how many features were measured in the initial statement "we performed multi-dimensional profiling of the synaptic puncta in DA9" and some description of what types. This is important for understanding the overall method (and comparing to the later set of 76 metrics mentioned on p.5) and shouldn't be buried in supp data.

Changes have been made as suggested.

- p.4 "metrics that fully capture phenotypes" mistakenly implies it is possible to truly fully

capture a phenotype using image-based metrics. Similarly, p. 5 "we thus conducted comprehensive phenotypic profiling" - the use of "comprehensive" is inaccurate.

Details explaining the fact that we are only referring to image-based phenotypes have been included.

- replace phenotypical with phenotypic

Changes have been made as suggested.

- p. 5 "cannot be identified from phenotyping a single animal" refers to reference 26 but it is not obvious how that reference supports the statement; it needs some elaboration (did that paper neglected to identify that mutant? did the paper do a comprehensive analysis of how many animals need to be examined to identify the phenotype?)

Agreed; changes have been made, along with additional explanation.

- p. 9 the main text needs more description of how the artificial images are constructed. In fact, I believe the term "schematic" would be more appropriate than "image" because the depictions are very simplified views of the phenotypes and are not at all similar to synthetic images (based on a full model of cell/organism appearance) that are sometimes produced in this field.

Agreed; changes have been made.

The term "can be represented in a visual manner" implies that some features cannot be so represented, but again it is indeed possible to construct synthetic images from raw measurements (see the work of the Bob Murphy lab, for example). So "can easily be represented" is more accurate.

Agreed; changes have been made.

- p. 9 the description of Fig S9 is not informative; it doesn't explain what sort of representation is presented. The supplement does not have a legend for a Fig S9 (my guess is the legend for S8 is the one meant, so there is a supp figure missing).

The legend for Figure S9 was mislabeled; it has been fixed.

- no reference to figures/data is provided for the statement "We subsequently tested this hypothesis and indeed found that a175 is an allele of lin-44 by complementation tests."

Details added, since a175 is so easily identifiable, this test was done by visual inspection

- p. 11 "is extremely subtle" > "are extremely subtle"

Thank you; changes have been made.

- Fig 1 b and c are not very explanatory; perhaps they need more text to explain (e.g. replace human head with "human-scored phenotype"), or should be omitted.

Figure 1b and 1c now include additional details to better explain the graphical representations, as suggested by the reviewer. These are meant to better illustrate the fact that quantitative multi-parametric analysis is better than qualitative visual inspection at identifying subtle phenotypes.

- Fig 2c shows 'relevant features' designated by numbers; the names of the features should be in the legend or on the plot.

We agree that the names of the relevant features should be displayed for those that are plotted. The names of the relevant features plotted in 2c have been included in the legend of the figure. Additional details for the remaining figures, plotted in Figure 2d are included in supplemental materials, and this is also noted in the legend of Figure 2.

- Fig 5c legend needs to explain what the size of each dot represents and explain the gray/green boxes at the lower left.

Details have been added to Figure 5c legend.

- Fig 5e legend needs to explain what the colors represent

Details have been added to Figure 5e legend, colored nodes are those where genetic relationships are also identified in the phenotypic similarity network analysis.

- Fig 5a legend needs to explain how samples are ordered, left to right

Details have been added to Figure 5a legend, samples are ordered in the same manner as Figure 3 for consistency. Strains are ordered by groups (known mutants, known mutants – ventral mislocalization, known mutants – not-visually identifiable, and mutants isolated in this screen). In each group, strains are ranked from most identifiable to least identifiable, based on the AUC metric. Details have also been added to Figure 3.